# Unlocking Speech–Text Compositional Powers: Instruction-Following Speech Language Models without Instruction Tuning

Congrui Du [* 1]   Yang Zhang [* 2]   Kaizhi Qian [* 2]   Shiyu Chang [1]

## Abstract

Instruction tuning for speech language models (SLMs) is substantially more challenging than for text-based large language models (LLMs), as it requires learning a new modality and a wide range of speech-specific instructions in addition to those supported by text LLMs. Existing SLM training approaches largely replicate the text LLM training paradigm by synthesizing large-scale speech pre-training and instruction-tuning datasets. However, this strategy is difficult to scale, since speech sequences are significantly longer than text sequences. In this paper, we propose SPEECHCOM-BINE, an instruction-following speech language model trained **without any instruction tuning**, using only a single round of speech pre-training on 30k hours of data. Starting from a text LLM base model, we perform continuous pre-training on speech utterances to obtain a speech-adapted model, and then directly combine its weights with the weight difference between the instruction-tuned and base versions of the text LLM. Our results show that this simple combination strategy not only preserves the knowledge and capabilities of the original text LLM, but also effectively transfers them to the speech domain. These findings suggest a new direction for SLM training that avoids reliance on massive speech data.

## 1. Introduction

Recent advances in large language models (LLMs) have catalyzed significant progress in speech language models (SLMs). One key capability of SLMs, as with text LLMs, is the ability to follow *instructions*. Unlike text LLMs, however, SLMs must handle a much broader spectrum of instructions. In particular, SLM instructions can be categorized into three types.

- **Text-oriented instructions.** These include instructions that standard text LLMs can already follow, such as question answering (QA), reasoning, and article generation. The key difference is that, for SLMs, the input instructions may be conveyed in either text or speech, and the model's responses can likewise be produced in either modality.

- **Speech understanding instructions.** These include instructions that query information from input speech utterances, particularly paralinguistic attributes such as emotion or emphasis. For these instructions, the input must contain speech, while the output may be in either speech or text.

- **Speech generation instructions.** These include instructions that require generating speech utterances subject to specific constraints, such as emotion or emphasized words. For this category, the input requirements may be expressed in either speech or text, but the output must be speech.

In summary, instruction following in SLMs introduces two additional levels of complexity beyond those faced by text LLMs. First, an SLM must follow the same instructions as a text LLM, but in a new modality. Second, it must handle entirely new, speech-specific instructions that text LLMs cannot address. As a result, instruction tuning for SLMs is substantially more challenging than for text LLMs.

To perform instruction tuning for SLMs, most existing methods largely replicate the training pipeline used for text LLMs (Ding et al., 2025). Usually, an SLM is first pre-trained on a large speech corpus using a next-token prediction objective, and then fine-tuned on datasets containing various speech instructions using supervised fine-tuning (SFT) and/or reinforcement learning (RL) (Ghosh et al., 2026). In other paradigms, only speech instruction-tuning is performed (Zhang et al., 2023). The training datasets typically consist of discrete speech tokens derived from real speech and/or synthetic speech generated from text datasets originally used for text LLM training.

However, this paradigm is heavily bottlenecked by the *data inflation* problem. For example, the sentence *"How are you"* comprises no more than five tokens in text form, whereas the

---

[1]University of California, Santa Barbara, USA [2]MIT-IBM Computing Research Lab, IBM Research, USA. Correspondence to: Yang Zhang <yang.zhang2@ibm.com>.

*Proceedings of the $43^{rd}$ International Conference on Machine Learning*, Seoul, South Korea. PMLR 306, 2026. Copyright 2026 by the author(s).

corresponding speech utterance typically lasts around one second and expands to roughly 60–200 speech tokens (Wang et al., 2025b), representing a $\sim 20\times$ increase relative to text. [1] Such inflation makes it significantly more difficult to scale the effective dataset size for SLM training. As a result, compared to text LLM training, SLM training must solve a substantially more challenging problem using far less effective data, which can lead to compromises in both knowledge acquisition and instruction-following capabilities.

More importantly, although SLM training is often initialized from a text LLM, subsequent training on massive speech tokens can lead to severe *catastrophic forgetting*. As a result, a substantial portion of SLM training is devoted to re-learning knowledge and skills in the original text LLM, often in a much harder-to-learn speech representation. If the knowledge and capabilities of the text LLM could be better preserved, SLM training would not need to be conducted at such an extensive scale. Some efforts (Wang et al., 2024) attempt to address this issue by freezing the text LLM and training only adapters for speech input and output. However, the resulting SLMs can only support text-oriented instructions, but not the other speech-related instructions.

In summary, existing SLM training paradigms involve an inherent trade-off between preserving the knowledge and capabilities of text LLMs and acquiring new, speech-specific skills. This raises a natural question: *Is it possible to achieve both objectives and thereby build more powerful SLMs?*

In this paper, we propose SPEECHCOMBINE, an SLM with strong instruction-following capabilities across all three instruction categories, whose training procedure is surprisingly simple: no iterative rounds of training, but only **a single round of pre-training** with a standard next-token prediction objective on just 30k hours of speech data. The core idea behind SPEECHCOMBINE is to directly transfer the instruction-following capabilities of text LLMs to a new modality and new skills, inspired by recent advances in model merging (Huang et al., 2024).

Figure 1 illustrates the basic idea. Given a text LLM base model and its instruction-tuned counterpart, we first compute their parameter difference, denoted as $\Delta\boldsymbol{\theta}_{inst}$ (blue arrow), which can be interpreted as a direction encoding instruction-following capability. Starting again from the base model, we then perform continuous pre-training on speech data, yielding a second weight difference $\Delta\boldsymbol{\theta}_{speech}$ (black arrow) that captures knowledge of the speech modality. Finally, SPEECHCOMBINE is constructed by transplanting $\Delta\boldsymbol{\theta}_{inst}$ onto the speech-adapted model (red arrow), based on the hypothesis that instruction-following capabilities can generalize to the speedh domain. In essence, our ap-

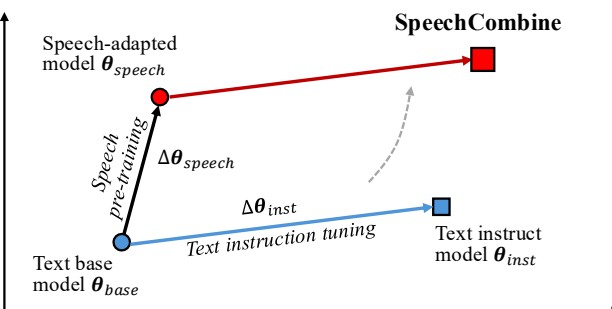

*Figure 1.* Illustration of SPEECHCOMBINE in weight space.

proach combines the speech adaptation direction $\Delta\boldsymbol{\theta}_{speech}$ with the instruction-following direction $\Delta\boldsymbol{\theta}_{inst}$ in parameter space, hence the name SPEECHCOMBINE.

Our experiments show that this combination unleashes surprising compositional effects, enabling the model to follow instructions across all three categories. The ability to follow speech understanding and generation instructions is particularly striking because neither $\Delta\boldsymbol{\theta}_{speech}$ nor $\Delta\boldsymbol{\theta}_{instr}$ has been exposed to these instructions during training. Furthermore, we find that this combination strategy also transfers other capabilities of the text LLM to the speech domain, such as *long-thinking capabilities*. These findings open up a new possibility for SLM training, escaping from the undue ordeal of inefficient data piling in today's world.

## 2. Related Work

**Speech Language Models.** SLMs can be classified into two primary categories: native multimodal and modular-aligned models. Native multimodal models (Défossez et al., 2024; Xie & Wu, 2024; Chen et al., 2025; Ding et al., 2025; Zeng et al., 2024; Geng et al., 2025; Wu et al., 2025; Xu et al., 2025a; Ghosh et al., 2026) employ a large language model (LLM) as the core reasoning backbone and extend it to the speech domain via audio encoders and speech decoders. In contrast, modular-aligned models (Shao et al., 2025; Lu et al., 2025; Wang et al., 2024; Fathullah et al., 2024; Kang et al., 2024; Wang et al., 2025a) freeze the backbone LLM while training modality adapters to grant the LLM audio capabilities. These methods either require large-scale training or are confined to text-related instructions. Existing approaches typically rely on large-scale audio instruction turning data, which is often expensive at scale. We aim to bridge this gap.

**Expressive Speech Synthesis.** Recent expressive TTS is moving from label-based control toward open-vocabulary, natural-language instructions for steering paralinguistic attributes. HiStyle (Zhang et al., 2025) improves text-to-style alignment with a hierarchical style predictor, while ParaStyleTTS (Lou et al., 2025) enhances robust paralinguistic control without reference audio. OV-InstructTTS (Ren

---

[1] Some speech tokenizers (Zeng et al., 2024) have a code rate of 25Hz, but each speech code typically consists of 8 tokens.

et al., 2026) and FlexiVoice (Chen et al., 2026) further strengthen instruction-following for expressive/zero-shot settings, the latter leveraging preference optimization to disentangle content, timbre, and style. Beyond single utterances, DeepDubbing (Dai et al., 2025) uses contextual InstructTTS for consistent audiobook narration.

**Speech Tokenizers.** Speech tokenizers can be categorized into *discrete* or *continuous* tokenization. Recent discrete tokenizers emphasize low bitrates and enhanced feature factorization: TaDiCodec (Wang et al., 2025b) employs a text-aware diffusion approach for ultra-low-rate modeling, while LongCat-Audio-Codec (Zhao et al., 2025) offers a streaming-friendly tokenizer-detokenizer solution tailored for practical SLMs. To improve generation quality, DSA-Tokenizer (Zhao et al.) disentangles semantic and acoustic features through hierarchical fusion. Alternatively, ProsodyLM (Qian et al., 2025) represents speech as transcripts augmented with explicit word-level prosody tokens. Several recent works (Yang et al., 2025b; He et al., 2025; Li et al., 2025) explore *continuous* tokenization. and aim to bypass the inherent information loss of quantization.

**Model Merging.** Model merging aims to integrate multiple capabilities within a single parameter space by combining model weights. Early works such as Task Arithmetic (Ilharco et al., 2022) and Model Soups (Wortsman et al., 2022) demonstrate that simple linear combinations of fine-tuned models can yield robust multi-task performance. Subsequent studies, including Chat Vector (Huang et al., 2024), BILLY (Pai et al., 2025), and Preference Vector (Liang et al., 2025), further show that task- or alignment-specific behaviors can be represented as weight-difference vectors and transferred across models without additional training. Model merging has also been explored as a strategy to mitigate catastrophic forgetting in multimodal adaptation (Chen et al., 2025; Sokar et al., 2025; Yu & Ananiadou, 2025). FunAudio-Chat (Chen et al., 2025) adopts the model merging idea in speech SLM, but it does not utilize the text LLM base model, which is shown to yield a more robust foundation for model merging (Anonymous, 2025).

## 3. Method

In this section, we describe how SPEECHCOMBINE is trained. Our goal is to retain the knowledge and instruction-following capabilities of a text LLM, while teaching the model to follow speech-related instructions.

### 3.1. The SPEECHCOMBINE Framework

Figure 2 illustrates the overall framework of SPEECHCOMBINE. Speech inputs are first converted into discrete speech tokens by the encoder module and then fed into the LLM. The LLM's generated outputs, also in the form of speech

tokens, are subsequently converted back into speech by the decoder. In this paper, we focus exclusively on training the LLM module. The encoder and decoder are instantiated using a pre-trained, open-source system (Section 3.5) and are kept frozen throughout training.

Section 1 and Figure 1 have already provided a brief overview of our weight combination framework for the LLM module. Here, we present a more formal description. Let $\boldsymbol{\theta}_{base}$ and $\boldsymbol{\theta}_{inst}$ denote the parameters of a text LLM base model and instruct model, respectively. Starting from the *base model*, we perform continuous pre-training on a speech corpus, yielding parameters $\boldsymbol{\theta}_{speech}$. We then define two directions in parameter space:

$$\Delta\boldsymbol{\theta}_{inst} = \boldsymbol{\theta}_{inst} - \boldsymbol{\theta}_{base}, \ \ \Delta\boldsymbol{\theta}_{speech} = \boldsymbol{\theta}_{speech} - \boldsymbol{\theta}_{base}. \quad (1)$$

$\Delta\boldsymbol{\theta}_{inst}$ and $\Delta\boldsymbol{\theta}_{speech}$ contain two complementary capabilities: $\Delta\boldsymbol{\theta}_{inst}$ encodes instruction-following behavior but contains no knowledge of speech, while $\Delta\boldsymbol{\theta}_{speech}$ captures knowledge of the speech modality but lacks the ability to follow natural language instructions. Our SPEECHCOMBINE model, parameterized by $\boldsymbol{\theta}_{SC}$, is obtained by linearly combining these two directions, *i.e.*,

$$\boldsymbol{\theta}_{SC} = \boldsymbol{\theta}_{base} + \lambda\Delta\boldsymbol{\theta}_{speech} + \Delta\boldsymbol{\theta}_{inst}, \quad (2)$$

where $\lambda \in (0, 1]$ is a soft weight.

One way to interpret the above combination is as follows. The instruction-following capabilities encoded in $\Delta\boldsymbol{\theta}_{inst}$ are tailored to the knowledge present in the base model. Equation (2) can thus be viewed as transferring $\Delta\boldsymbol{\theta}_{inst}$ onto the speech-adapted model, $\boldsymbol{\theta}_{base} + \lambda\Delta\boldsymbol{\theta}_{speech}$, which incorporates new knowledge of speech. The success of this transfer depends on how close the speech-adapted model remains to the original base model. A smaller value of $\lambda$ keeps the speech-adapted model closer to the base model, making it easier for $\Delta\boldsymbol{\theta}_{inst}$ to generalize. However, choosing $\lambda$ too small would limit the acquisition of speech knowledge. Therefore, this trade-off must be carefully balanced.

One important caveat is that the continuous pre-training on speech must start from the base model $\boldsymbol{\theta}_{base}$ rather than the instruction-tuned model $\boldsymbol{\theta}_{inst}$, which is a major distinction from existing SLM training recipes that also involve model merging (Chen et al., 2025). Training from base is crucial in retaining the compositional structure of knowledge and skill inside the SLM to allow for a successful combination.

Since the training of SPEECHCOMBINE only involves one round of pre-training, the algorithm design boils down to designing a good recipe for the continuous pre-training, which we detail in Sections 3.2 to 3.5.

### 3.2. Knowledge of Speech

Before introducing the details of speech pre-training (which aims to learn speech knowledge), it is important to explain

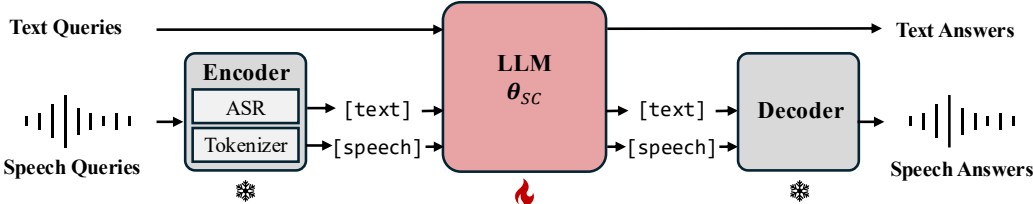

*Figure 2.* The SPEECHCOMBINE framework and inference pipeline.

what speech knowledge is. Speech carries multiple levels of information, including ❶ **content**, which can typically be transcribed into text; ❷ **prosody**, which describes properties such as pitch, intonation, rhythm, and loudness; and ❸ **timbre**, which characterizes the speaker's voice. Since the majority of the speech-related instructions are about the prosodic aspect of speech, such as expressive generation, word emphasis detection, *etc.*, we limit ourselves to the first two levels of information in this paper. Future generalization to timbre is easily achievable in our framework.

The speech modality differs from text in two fundamental ways. First, speech token sequences differ substantially from text token sequences in their structure and vocabulary. Second, speech carries much richer information than text. As a result, during pre-training, an SLM must achieve two goals: ❶ **become familiar with speech token sequences**, and ❷ **understand the information in speech** (content and prosody in this paper). Section 3.3 describes how we design the pre-training scheme to achieve these two goals.

### 3.3. Pre-training Data Structure

We adopt the standard next-token prediction objective for pre-training. As a result, the key design choice in our pre-training recipe lies in constructing training data that can achieve the two goals described above under the next-token prediction regime. To this end, we design the following pre-training data structure:

```
[S1 cap] [S1 text] [S1 speech] [S1 cap]
[S2 cap] [S2 text] [S2 speech] [S2 cap] ···
```

'S1' stands for sentence 1; 'speech' stands for speech token sequence; 'cap' stands for speech caption.

The speech captions describe the information conveyed in speech, including both content and prosody, in natural language, so the [cap] section is essential for achieving the second goal outlined in Section 3.2. Each [speech] section contains the speech token sequence for a single sentence and familiarizes the SLM with the structure of speech token sequences, thereby addressing the first goal. Sections 3.4 and 3.5 detail our designs of the speech caption and speech token sequence, respectively.

During both pre-training and inference, each [speech] section is always preceded by a [text] section, which con-

tains the *exact transcription* of the corresponding speech tokens. Although this design introduces some redundancy, we show that it is necessary to facilitate the transfer of text-based instruction-following capabilities to speech (Section 3.6). Section 3.5 describes how we reduce this redundancy through speech tokenization.

The [cap] section may appear before or after [speech]. We adopt a randomization strategy: for each sentence, [cap] appears before [speech] with probability 0.3, appears after [speech] with probability 0.3, and is omitted with probability 0.4. This ensures that the pre-training data contain sufficient instances of the following three transition patterns: ❶ [cap][text][speech], which teaches knowledge for speech generation; ❷ [text][speech] [cap], which teaches knowledge for speech understanding; and ❸ [text][speech][text][speech], which teaches knowledge for multi-sentence speech generation. Other pre-training details can be found in Appendix A.1.

### 3.4. Speech Caption

To generate a caption for a speech utterance, we first extract the following attributes from the utterance: average pitch level (high, medium, or low), average speaking rate (high, medium, or low), emphasized words, perceived emotion, emotional arousal (i.e., the intensity of the emotion), and the text transcription. We then randomly select a subset of these attributes to include in the caption: specifically, we sample an integer $k$ uniformly from 1 to the total number of available attributes for the sentence and include $k$ attributes in the caption. Finally, the selected attributes are provided to GPT-OSS 120B (Agarwal et al., 2025), which is prompted to generate a natural-language description that covers all supplied attributes. Below is an example generated caption:

<cap> *The spoken excerpt is a read passage rendered at a medium tempo, featuring a high pitch, spotlighting the word "nose", and expressing anger.*</cap>

Additional details are provided in Appendix A.2.

### 3.5. Speech Tokenization

To reduce redundancy between the [text]–[speech] pairs, we aim to adopt a tokenization scheme whose encoded information has minimal overlap with the text transcription.

To this end, we adopt the prosody tokenization scheme proposed in Qian et al. (2025), which encodes only prosodic information in speech and therefore produces substantially shorter token sequences than schemes that encode the full waveform. Specifically, for each word in a speech utterance, the prosody tokenization scheme generates five numerical values describing the word's median pitch, pitch range, pitch slope, duration, and energy. These values are quantized into 512 discrete levels, yielding a sequence of discrete *prosody tokens*. We introduce delimiter tokens `<speech>` and `</speech>` to enclose each speech token sequence. Additional details are provided in Appendix A.3.

### 3.6. Inference

Once the model has been pre-trained and the instruction direction has been integrated according to Equation (2), the resulting SPEECHCOMBINE is ready to answer real user queries. This naturally raises our next research questions: What input–output format should be used during inference? Can SPEECHCOMBINE, which is never trained on the inference tasks themselves, reliably follow this format?

The following shows our inference template, based on a model derived from QWEN3-8B (Yang et al., 2025a).

```
<|im_start|> User
What is the emotion of the following
utterance?[text][speech]<|im_end|>

<|im_start|> Assistant
[text][speech][text][speech]...<|im_end|>
```

As can be observed, our answer template is the same as that of the corresponding LLM instruct model, except that it includes `[speech]` sections to enable speech-mode interaction. Since each `[speech]` section is always accompanied by its corresponding `[text]` section (as discussed in Section 3.3), SPEECHCOMBINE relies on an external ASR system to transcribe an input speech query into the `[text]` section before prosody tokens are extracted, as illustrated in Figure 2. We leave internalizing the ASR module within the model as a direction for future work.

The answer template above also clarifies why SPEECHCOM-BINE, despite not being trained on this inference format, is still able to produce correct responses. This is because the instruction-following direction of the text LLM ($\Delta\boldsymbol{\theta}_{inst}$) already recognizes most of the template structure. The only additional element, the `[speech]` section, is readily handled by the speech pre-training direction ($\Delta\boldsymbol{\theta}_{speech}$). The `[text]` sections serve as anchors for transferring the text LLM's capabilities, which is why the `[text]` and `[speech]` sections must always appear in pairs.

To further improve the stability of SPEECHCOMBINE following this format, we introduce *format forcing*. Specif-

ically, when generating a `[text]` section, we mask out the speech-token vocabulary to prevent unintended speech outputs. When speech mode is enabled, we boost the generation probability of the delimiter `<speech>` to ensure that a speech section is produced at the end of each text section. Additional inference details are provided in Appendix A.4.

### 3.7. Enabling Long Thinking

Section 3.6 shows that, as long as the answer template closely matches that of the text LLM, the instruction-following abilities of the instruct model ($\boldsymbol{\theta}_{inst}$) can be readily transferred to SPEECHCOMBINE. We thus wonder: Can other abilities in $\boldsymbol{\theta}_{inst}$ be transferred as well?

We thus seek to elicit another prominent capability of many text LLMs – *long thinking*, by modifying the answer template in the same manner used to invoke long thinking in text LLMs. For example, when the text LLM is QWEN3-8B, we simply insert a `<think> [text]</think>` section immediately after `Assistant`. This is implemented by forcing the first token after `Assistant` to be `<think>`. Additional design choices that stabilize the onset of long-thinking generation are discussed in Appendix A.5.

Unlike other SLMs, SPEECHCOMBINE acquires long thinking almost *'for free'* – without any modification to the training procedure or weight combination, but simply through a change in the answer template. Accordingly, we enable the long-thinking mode by default in this paper.

## 4. Experiments

In this section, we present experiments to evaluate the effectiveness of SPEECHCOMBINE.[2] Unlike most existing work in this area, our approach is not directly trained on any of the evaluation tasks. We divide the evaluation tasks into two categories based on the required levels of skill combination:

**Shallow combination.** This category includes text-oriented tasks that the text LLM can already perform, but posed and answered in the *speech* modality. These tasks primarily require combining the problem-solving capabilities in $\Delta\boldsymbol{\theta}_{inst}$ with the speech input–output format learned in $\Delta\boldsymbol{\theta}_{speech}$, which is relatively straightforward.

**Deep combination.** This category includes speech-related tasks, such as speech generation and speech understanding, that are completely unseen during training. These tasks require transferring the instruction-following capabilities in $\Delta\boldsymbol{\theta}_{inst}$ to the newly acquired speech knowledge in $\Delta\boldsymbol{\theta}_{speech}$, which is much more challenging.

---

[2]Github: https://github.com/CongruiDu/SpeechCombine
Demo Webpage: https://auspicious3000.github.io/SpeechCombine-Demo

Table 1. Results on text-oriented tasks. The metric is accuracy (%) ↑ for all the tasks. Among all the methods in **Group B**, except for GPT-4O-AUDIO with significantly larger model size, the best results are noted in bold, second best results underlined.

| Methods | (a) **QA** | | (b) **Reasoning** | | | |
| --- | --- | --- | --- | --- | --- | --- |
| | OpenbookQA | MMSU | GSM8k | Truthful | MLC | MLCpro |
| GPT-4O-AUDIO[1,2] | 89.23 | 80.25 | 80.00 | 82.67 | 80.00 | 46.67 |
| ASR + TEXT LLM | 83.29 | 73.22 | 94.61 | 71.12 | 93.26 | 94.13 |
| CONT. PRE-TRAIN | 78.46 | 68.21 | 87.05 | 42.11 | 85.31 | 88.27 |
| CONT. PRE-TRAIN + SFT | 80.21 | 60.8 | 87.34 | 42.58 | 83.23 | 88.27 |
| GLM-4-VOICE[1,2] | 53.41 | 39.75 | 30.93 | 59.28 | 57.82 | 65.20 |
| AUDIO-FLAMINGO 3 | 58.68 | 42.19 | 37.11 | 32.02 | 68.36 | 61.9 |
| STEP-AUDIO2-MINI | 72.74 | 54.42 | 42.89 | 53.87 | 87.38 | 80.21 |
| STEP-AUDIO2-MINI-THINK | 65.70 | 53.87 | 39.46 | 52.89 | 85.12 | 76.55 |
| OSUM-ECHAT[3] | 78.46 | 60.11 | 32.41 | 36.82 | 46.70 | 51.28 |
| QWEN-2.5-OMNI | 81.1 | 61.32 | 38.09 | 46.25 | 73.33 | 64.83 |
| KIMI-AUDIO[1] | _83.52_ | 62.17 | **95.53** | 57.61 | 93.59 | 87.91 |
| FUN-AUDIO-CHAT[4] | _83.52_ | _71.08_ | 88.31 | **61.27** | **93.97** | **93.40** |
| **SPEECHCOMBINE** | **86.59** | **73.38** | _90.03_ | _60.09_ | **93.97** | _89.01_ |

1. QA results copied from Chen et al. (2024).  2. Reasoning results copied from Yan et al. (2025).
3. QA results copied from Geng et al. (2025).  4. QA results copied from Chen et al. (2025).

We will first present results on shallow combination in Section 4.2, then deep combination in Sections 4.3 and 4.4.

## 4.1. Experimental Configurations

We choose the QWEN3-8B-base and -instruct as our $\theta_{base}$ and $\theta_{inst}$, respectively. For the continuous pre-training on speech, we apply LoRA-based tuning (Hu et al., 2022) with rank 64 and $\alpha$ 16. The pre-training datasets include Librilight (Kahn et al., 2020), BEAT (Liu et al., 2022), CREMA-D (Cao et al., 2014), ESD (Zhou et al., 2022), JL Corpus (James et al., 2018), EmoV-DB (Adigwe et al., 2018), Expresso (Nguyen et al., 2023), MEAD (Wang et al., 2020) TESS (Pichora-Fuller & Dupuis, 2020). We follow Qian et al. (2025) to extract the prosody tokens and use whisper-large-v3 (Radford et al., 2023) to extract text transcriptions, and the total amount of data, after dropping extration failures, is only around 30k hours. $\lambda$ is set to 0.85.

We include two groups of baselines. **Group A** consists of alternative training methods using comparable training data and the same base models as for SPEECHCOMBINE. This group is more controlled to better study the effectiveness of SPEECHCOMBINE compared to the common SLM training paradigms. It includes -

• ASR + TEXT LLM: Input speech is converted to text using whisper-large-v3 and then fed to the QWEN3-8B-instruct model;

• CONTINUOUS PRE-TRAIN: Continuous pre-training on the QWEN3-8B-instruct model (instead of base), using the same 30k hours of speech data;

• CONTINUOUS PRE-TRAIN + SFT: Continuous pre-training on the QWEN3-8B-base model, using the same 30k hours of speech data, and then further fine-tuning on

around 10,000 hours of speech instruction-tuning data from SIFT-50M (Pandey et al., 2025), InstructS2S-200K (Fang et al., 2025), and VoiceAssistant-400K (Xie & Wu, 2024), using SFT;

**Group B** consists of state-of-the-art SLMs, including: GPT-4O-AUDIO (OpenAI, 2024), GLM-4-VOICE (9B) (Zeng et al., 2024), AUDIO FLAMINGO 3 (Ghosh et al., 2026), STEP-AUDIO2-MINI (7B) (Wu et al., 2025), STEP-AUDIO2-MINI-THINK (7B) (Wu et al., 2025), OSUM-ECHAT (3B) (Geng et al., 2025), QWEN-2.5-OMNI (Xu et al., 2025b), KIMI-AUDIO (7B) (Ding et al., 2025), and FUN-AUDIO-CHAT (8B) (Chen et al., 2025). All baselines undergo multiple rounds of training on substantially larger datasets than SPEECHCOMBINE. For example, FUN-AUDIO-CHAT was trained on millions of hours of speech data (Chen et al., 2025), over 100× the size of our training data. All the models are of similar sizes, except for GPT-4O-AUDIO, whose results are thus excluded from bolding.

## 4.2. Text-Oriented Tasks

We consider two text-oriented tasks: QA and logical reasoning. QA is included because it represents a common application for SLM, while logical reasoning is used to challenge the models with more complex questions. For QA, we evaluate on OpenbookQA, SDQA, and MMSU in VoiceBench (Chen et al., 2024). For logical reasoning, we evaluate on GSM8kEval, MLCpro-en, TruthfulEval, and MLC in URO-Bench (Yan et al., 2025). Additional details are provided in Appendix B.1.

Table 1 summarizes the results. Among the Group A methods, ASR + TEXTLLM represents the topline performance, because it does not suffer from any catastrophic forgetting. Nevertheless, SPEECHCOMBINE performs on par or even

*Table 2.* Results on speech understanding and generation tasks. Among all the methods in **Group B**, except for GPT-4O-AUDIO with significantly larger model size, the best results are noted in bold, second best results underlined.

| Methods | (a) **Speech Understanding** | | | | (b) **Speech Generation** | | | |
| | UnderEmo | Emph Detection | | | GenEmo | Emph Generation | | |
| | Acc. ↑ | Prec. ↑ | Recall ↑ | F1 ↑ | Score ↑ | Prec. ↑ | Recall ↑ | F1 ↑ |
|---|---|---|---|---|---|---|---|---|
| GPT-4O-AUDIO[1] | 48.53 | 33.00 | 61.68 | 42.99 | 33.46 | 66.95 | 63.19 | 65.02 |
| ASR + TEXT LLM[2] | 55.42 | 15.79 | 26.94 | 19.91 | 5.06 | 11.35 | 29.71 | 16.42 |
| CONT. PRE-TRAIN | 47.15 | 1.53 | 6.51 | 2.48 | 24.23 | 6.9 | 17.5 | 9.89 |
| CONT. PRE-TRAIN + SFT | 48.90 | 2.12 | 3.42 | 2.61 | 24.23 | 16.76 | 26.95 | 20.67 |
| GLM-4-VOICE | 52.41 | 11.92 | 29.52 | 16.98 | **48.13** | _24.67_ | _28.75_ | _26.55_ |
| AUDIO-FLAMINGO 3[3] | 24.28 | 17.09 | 33.32 | 22.59 | - | - | - | - |
| STEP-AUDIO2-MINI | 44.57 | 19.59 | 25.76 | 22.25 | 31.35 | 22.48 | 23.23 | 22.85 |
| STEP-AUDIO2-MINI-THINK | 45.83 | 20.37 | _39.45_ | 26.87 | 23.73 | 22.75 | 23.11 | 22.93 |
| QWEN-2.5-OMNI | 35.86 | 3.75 | 26.26 | 6.56 | 13.54 | 18.82 | 34.03 | 24.24 |
| OSUM-ECHAT | 48.71 | 22.51 | 9.02 | 12.88 | 36.01 | 13.87 | 19.61 | 16.25 |
| KIMI-AUDIO | _63.69_ | 19.71 | 20.04 | 19.88 | 28.01 | 18.21 | 26.81 | 21.69 |
| FUN-AUDIO-CHAT | **74.74** | _23.46_ | 37.15 | _28.76_ | 39.30 | 23.01 | 22.82 | 22.91 |
| **SPEECHCOMBINE** | 52.70 | **55.11** | **67.89** | **60.84** | _45.42_ | **25.91** | **39.90** | **31.42** |

1. `UnderEmo` and `GenEmo` results copied from Yan et al. (2025).
2. Speech generation achieved by appending `Kokoro` text-to-speech (hexgrad, 2025).
3. Generation task results unavailable because the audio decoder has not been released.

better compared with this topline. With additional training, the other two methods in Group A yield much compromised performance, underperforming SPEECHCOMBINE. However, the catastrophic forgetting is not as severe as we anticipated, which we suspect is due to the small dataset size for the additional training. As shown in Sections 4.3 and 4.4, although Group A methods decently retain text LLM's knowledge, it is at the cost of severe failures in speech-related tasks, reflecting difficult trade-offs in these conventional training paradigms.

Among the Group B methods with comparable model sizes, SPEECHCOMBINE achieves either the top or second-best results across all datasets. This performance is particularly notable given that the top-performing SLM baselines are trained on extensive math and reasoning datasets to strengthen these capabilities (Ding et al., 2025), whereas SPEECHCOMBINE is not. These results demonstrate that our weight combination strategy successfully preserves the original capabilities of the text LLM, while seamlessly integrating speech-mode interaction without interfering with correct content generation.

### 4.3. Speech Understanding Tasks

We consider two speech understanding tasks: emotion understanding and emphasis detection. Emotion understanding requires identifying the speaker's emotion based on prosodic cues in an utterance. We evaluate this task on the `UnderEMO-en` benchmark from `URO-Bench`. Emphasis detection aims to identify words in an utterance that are emphasized by the speaker. We adapt the `EmphAssess` dataset (de Seyssel et al., 2024), which was originally a

benchmark for mimicking emphasis patterns from a reference utterance. We treat the reference utterance as the input speech and use the provided emphasis annotations as ground truth. Performance is evaluated using precision, recall, and F1 score computed over the predicted emphasized words. Additional details are provided in Appendix B.2.

Table 2 (Panel a) reports the results. As shown, SPEECH-COMBINE does not achieve top-tier performance on emotion understanding. This is likely due to the insufficient emotion information in the speech captions: only about 100 hours of the pre-training corpus include emotion captions. In contrast, more than 15k hours contain emphasis-related information. As a result, SPEECHCOMBINE achieves state-of-the-art performance on the emphasis detection task, with substantial margins over all baselines. Group A methods, with no or insufficient adaptation to speech, fail in speech understanding tasks, especially for emphasis detection. Yet their performance does not drop to zero because they can still partially infer the answers based on text only.

One important caveat is that speech understanding capabilities are inherently more difficult to elicit than shallow combination capabilities. In practice, we find that SPEECHCOMBINE tends to answer such questions primarily based on textual content, rather than prosodic cues, unless additional measures are taken to explicitly encourage prosody-aware reasoning. For example, in the emphasis detection task, we prepend *"Based on the prosody"* to the long-thinking section to force the model to attend to prosodic information. Section 4.5 provides a more detailed analysis of this behavior. This design choice introduces a mild advantage over baseline methods. Nevertheless, the strong performance of

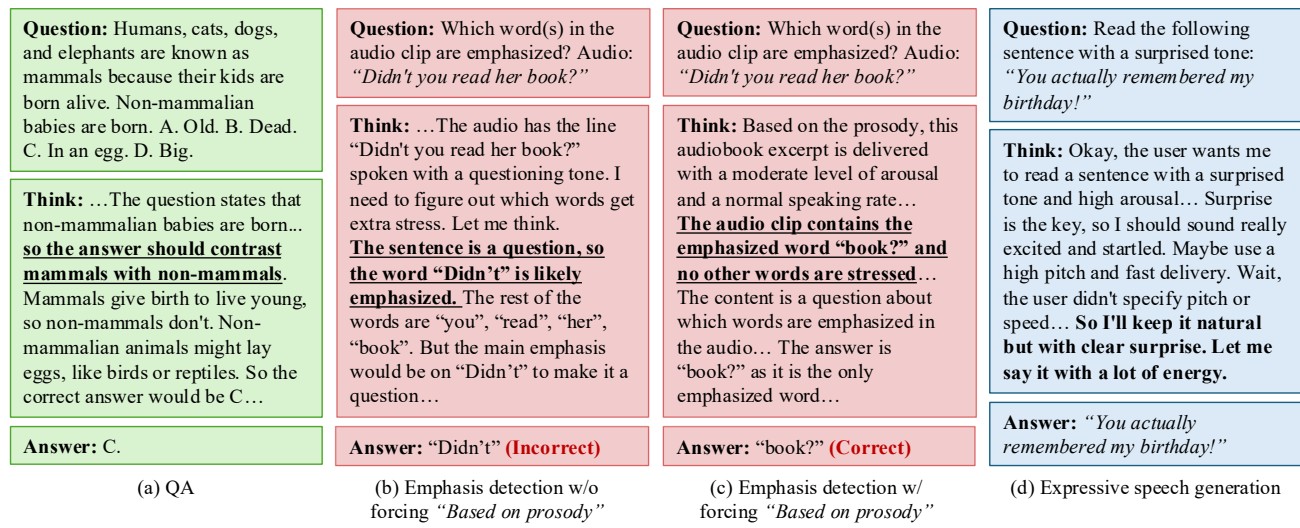

| Question: Humans, cats, dogs, and elephants are known as mammals because their kids are born alive. Non-mammalian babies are born. A. Old. B. Dead. C. In an egg. D. Big. | Question: Which word(s) in the audio clip are emphasized? Audio: *"Didn't you read her book?"* | Question: Which word(s) in the audio clip are emphasized? Audio: *"Didn't you read her book?"* | Question: Read the following sentence with a surprised tone: *"You actually remembered my birthday!"* |
| Think: …The question states that non-mammalian babies are born… **so the answer should contrast mammals with non-mammals**. Mammals give birth to live young, so non-mammals don't. Non-mammalian animals might lay eggs, like birds or reptiles. So the correct answer would be C… | Think: …The audio has the line "Didn't you read her book?" spoken with a questioning tone. I need to figure out which words get extra stress. Let me think. **The sentence is a question, so the word "Didn't" is likely emphasized.** The rest of the words are "you", "read", "her", "book". But the main emphasis would be on "Didn't" to make it a question… | Think: Based on the prosody, this audiobook excerpt is delivered with a moderate level of arousal and a normal speaking rate… **The audio clip contains the emphasized word "book?" and no other words are stressed**… The content is a question about which words are emphasized in the audio… The answer is "book?" as it is the only emphasized word… | Think: Okay, the user wants me to read a sentence with a surprised tone and high arousal… Surprise is the key, so I should sound really excited and startled. Maybe use a high pitch and fast delivery. Wait, the user didn't specify pitch or speed… **So I'll keep it natural but with clear surprise. Let me say it with a lot of energy.** |
| Answer: C. | Answer: "Didn't" **(Incorrect)** | Answer: "book?" **(Correct)** | Answer: *"You actually remembered my birthday!"* |
| (a) QA | (b) Emphasis detection w/o forcing *"Based on prosody"* | (c) Emphasis detection w/ forcing *"Based on prosody"* | (d) Expressive speech generation |

*Figure 3.* Excerpts of thinking processes of SPEECHCOMBINE. Non-text sections are removed for readability.

SPEECHCOMBINE remains encouraging: It confirms that the weight combination strategy is capable of cultivating novel speech understanding capabilities, even though additional mechanisms are currently required to activate them.

## 4.4. Speech Generation Tasks

We consider two speech generation tasks: expressive speech generation and emphasis generation. Expressive speech generation requires producing speech utterances that satisfy given emotion constraints. We evaluate this task using the GenEmotion-en benchmark from URO-Bench. Performance is measured using a composite score that combines word error rate with the correctness of the expressed emotion, as determined by an external emotion classifier. Emphasis generation involves generating speech utterances that satisfy given emphasis requirements. For this task, we again adapt the EmphAssess dataset by converting emphasis labels into emphasis constraints, and we use the provided emphasis detector to evaluate whether the generated speech emphasizes the required words. Performance is reported using precision, recall, and F1 score. Additional details are provided in Appendix B.3.

Table 2 (Panel b) reports the results. Surprisingly, speech generation capabilities prove easier to elicit than speech understanding capabilities, and do not require any task-specific enforcement mechanisms. The resulting generation quality is strong: SPEECHCOMBINE achieves the second-best performance on emotion generation and the best performance on emphasis generation, with clear margins over the baselines. Group A methods, in particular, achieve the worst performance due to the insufficient acquisition of speech skills. We encourage readers to refer to our demo webpage to judge the perceptual quality of the generation.

## 4.5. Visualizing the Thinking Process

To gain further insight into how SPEECHCOMBINE leverages its combined capabilities, we select three representative tasks – OpenbookQA, emphasis detection on EmphAssess, and GenEmo – one from each task category. For each task, we visualize the model's reasoning process and answer for a representative example.

Figure 3(a) illustrates the thinking process for a QA example, which closely resembles the typical reasoning behavior of text LLMs. This observation further confirms that the original capabilities of the text LLM are well preserved.

Figure 3 (b) and (c) show the thinking process for the same emphasis detection question with (c) and without (b) prepending *"Based on the prosody"* to the thinking (Section 4.3). Interestingly, without the forcing, SPEECHCOMBINE attempts to infer what should be emphasized based on the text, which leads to an incorrect answer. Only when it is forced to focus on prosody does it start to reason what is actually being emphasized by the speaker, which leads to the correct answer. This contrast suggests that while weight combination enables the emergence of speech understanding capabilities, it is not sufficient on its own to reliably signal when and how these capabilities should be invoked.

Figure 3(d) shows the thinking process for an expressive speech generation task. Remarkably, the model is able to reason about the most appropriate prosodic styles to accentuate the requested emotion of excitement, exhibiting a previously unseen form of long-thinking behavior. Specifically, the model considers increasing pitch or speaking rate, and ultimately decides to increase energy. This observation provides strong evidence that long-thinking behavior has been extended to solve unseen, speech-centric tasks.

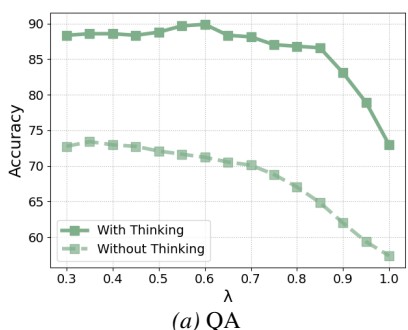
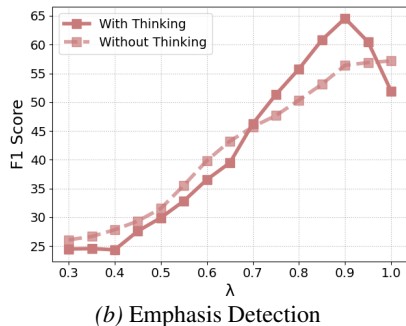
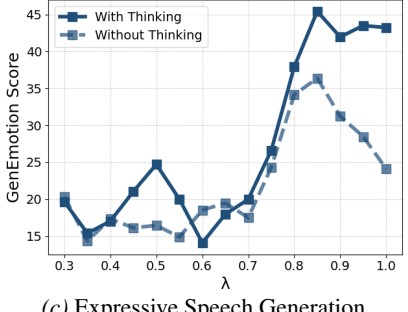

|(a) QA | (b) Emphasis Detection | (c) Expressive Speech Generation |

*Figure 4.* Performance across different $\lambda$.

*Table 3.* Ablation study results.

| Methods | Openbook Acc. ↑ | Emph Det F1 ↑ | GenEmo Score ↑ |
|---|---|---|---|
| Original | 86.59 | 60.84 | 45.42 |
| No Thinking | 64.83 | 53.14 | 36.37 |
| No [cap] | 84.83 | 0.39 | 27.18 |
| No [Text] | 83.95 | 0.40 | 35.69 |
| No $\Delta\boldsymbol{\theta}_{inst}$ | 71.68 | 40.38 | 15.34 |

### 4.6. Ablation Studies

**Combination weight.** As discussed in Section 3.1, the parameter $\lambda$ in Equation (2) controls the trade-off between acquiring speech-specific knowledge and preserving the transferability of text-based instruction-following capabilities. We examine how different choices of $\lambda$ affect performance, using the same three tasks considered in Section 4.5.

Figure 4 (solid curves) plots task performance as a function of $\lambda$, revealing clear contrasting trends. As $\lambda$ increases, performance on QA, a text-oriented task, degrades due to greater perturbations to the original text LLM weights, while performance on the two speech-related tasks improves as speech knowledge is more strongly integrated. Despite this trade-off, there exists a broad sweet spot for $\lambda$ that yields competitive, well-balanced performance across all tasks.

**Effect of long thinking.** We also plot the corresponding performance curves for SPEECHCOMBINE in non-thinking mode in Figure 4 (dashed curves). As shown, performance without long thinking is consistently lower, confirming that long thinking provides meaningful benefits even on unseen tasks. Nevertheless, the non-thinking model still achieves competitive performance, further validating that the strong results primarily stem from the effective skill combination enabled by our weight combination scheme.

**Pre-training data structure.** Our pre-training data contains three sections, [speech], [cap], [text]. The first is essential because it is the only section with speech information. To study whether the other two sections are necessary, we remove them from pre-training and evaluate the resulting models on the same three tasks. As shown in

Table 3, when the [cap] is removed, the speech generation and understanding performance degrades significantly, because [cap] introduces the knowledge of speech generation and understanding. When [text] is removed, the speech generation and understanding performance is also significantly impacted, because [text] serves as an anchor for fusing speech knowledge.

**Weight directions.** To study the effect of $\Delta\boldsymbol{\theta}_{inst}$, we remove $\Delta\boldsymbol{\theta}_{inst}$, but instead use task-specific in-context examples to provide instruction-following guidance. The results in Table 3 show that in-context learning can achieve better performance on the text-oriented task than the no-thinking SPEECHCOMBINE, but worse for speech understanding and generation, which suggests it cannot completely replace $\Delta\boldsymbol{\theta}_{inst}$. On the other hand, if $\Delta\boldsymbol{\theta}_{speech}$ is removed, the model becomes ASR + TEXT LLM (Tables 1 and 2), which excels in text-oriented tasks but fails in speech-related tasks.

More ablation studies are shown in Appendix C.

## 5. Conclusion and Limitations

In this paper, we explore weight combination as an alternative paradigm for training instruction-following SLMs that avoids multi-round training on massive data. The resulting empirical findings exceed our expectations: SPEECHCOMBINE achieves state-of-the-art performance across multiple tasks, with less than 1% of training data for existing SLMs.

Despite these promising results, SPEECHCOMBINE in its current form has several limitations. First, its output formatting is unstable, requiring format forcing for correction. Second, the prosody tokenization scheme used in this work does not encode timbre, voice quality, or accent information, which limits its applicability to a broader range of speech tasks. Finally, reliance on an external ASR system introduces additional latency and transcription errors during inference. Nevertheless, this paper unveils great potential of our proposed paradigm. Beyond enabling the development of competitive SLMs today, this paradigm provides a scalable path for SLMs to rapidly inherit emerging capabilities from text LLMs as they continue to evolve in the future.

## Impact Statement

This paper presents work whose goal is to advance the field of machine learning. There are many potential societal consequences of our work, none of which we feel must be specifically highlighted here.

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

# A. Additional Algorithm Details

## A.1. Speech Pre-training

**Special Tokens.** Certain special tokens play a crucial role during instruction-following, but they do not appear in the data for the speech continuous pre-training. As a result, SPEECHCOMBINE will forget the ability to output these special tokens during inference, leading to formatting failures. These tokens include `<|im_end|>` which marks the end of the answer, and `</think>`, which marks the outset of the long thinking section[3]. To prevent forgetting of these tokens, we randomly insert them at the end of each `[text]` section in our training data with probability $p$. $p$ is set to 0.2 for `<|im_end|>`, and 0.02 for `</think>`.

**Data Interleaving.** Our pre-training data are mostly audiobooks, which consist of very long segments of speech utterances by a single speaker, with consistent prosody styles. However, during inference, the model is applied to a user-agent interaction setting, which involves alternating speakers and speaking styles. To make the model aware of the speaker's turn-taking, we chop the audiobooks into small segments containing 5-7 sentences, and piece together small segments from different audiobooks. An alternative approach is to use conversational speech corpora instead of audiobooks, which we will explore as a future direction.

## A.2. Speech Caption

We describe how we derive each attribute below.

**Average Pitch Level.** We use RMVPE Wei et al. (2023) to extract the fundamental frequency ($F_0$), and quantize it into three pitch classes: `Low`, `Medium`, and `High`. Since pitch ranges vary substantially across speakers and genders, we define relative thresholds based on median pitch statistics at both the speaker and gender levels.

Specifically, we compute the median pitch $m_s$ for each speaker and the median pitch $m_g$ for each gender group, and apply the following criteria:

- `Low`: $F_0 < 0.75m_s$ (speaker-level) **or** $F_0 < 0.85m_g$ (gender-level),

- `High`: $F_0 > 1.30m_s$ (speaker-level) **or** $F_0 > 1.20m_g$ (gender-level),

- `Medium`: otherwise.

The final pitch class is determined by an OR-based fusion of the speaker-level and gender-level decisions, i.e., a frame is labeled as `Low` or `High` if either criterion is satisfied; otherwise, it is labeled as `Medium`.

**Average Speaking Rate.** Following Jin et al. (2024), we use the open-sourced Python package `g2p_en` to convert transcripts into phonemes and compute the average time per phoneme $Sr$. We then compute the median value $m$. For each utterance, we apply the following criteria:

- `Low`: $Sr > 1.3m$,

- `High`: $Sr < 0.85m$,

- `Medium`: otherwise.

**Emphasized Words.** We utilize Whistress Yosha et al. (2025) as our emphasized words extractor.

**Emotional arousal.** We use whisper-large-v3-msp-podcast-emotion-dim model introduce in Feng et al. (2025) as our arousal extractor. Due to the model only support audio input with length of 3 to 15 seconds, we adopt sliding window technique to compute the arousal score $A$ of any unqualified audio. For each utterance, we apply the following criteria:

- `Low`: $A < 0.282$,

- `High`: $A > 0.54$,

---

[3]Special tokens that mark the outset of answering sections do not produce as much negative impact, because they are always hard-coded in the answer template, rather than generated by the model.

- `Medium`: otherwise.

**Sampling Strategy.** During the caption generation phase, we uniformly sample one attribute from the full set of available attributes. If the sampled attribute is *text*, it is dropped with a probability of 90%. For emotion-related datasets, the *emotion* attribute is always guaranteed to be sampled.

## A.3. Prosody Tokenization

For better explanation, consider an example where the content of a speech utterance is *"How are you?"* The corresponding `[speech]` section takes the following format.

```
<speech><spk f0 med>
<SIL><Dur> how<Dur><f0 range><f0 med><f0 slope><energy>
<SIL><Dur> are<Dur><f0 range><f0 med><f0 slope><energy>
<SIL><Dur> you<Dur><f0 range><f0 med><f0 slope><energy>
<SIL><Dur> </speech>
```

where `<SIL>` represents the silence in between words. `<Dur>` represents the duration of the word (in frames) of the word or silence in between words. `<f0 range>` represents the the difference between the 95% and 5% percentiles of the frame-level Log-F0 contour of the word. `<f0 med>` represents the median of the frame-level Log-F0 contour of the word. `<f0 slope>` represents the slope of the straight line that best fits the Log-F0 contour of the word. `<energy>` represents the log-norm of the mel-spectrogram of the word. `<spk f0 med>` represents the median of the frame-level Log-F0 contour the speaker.

As can be seen, the speech token sequence consists of two alternating patterns: ❶ A `<SIL>` followed by a `<Dur>` which specifies the duration of the silence in between two words, or at the start/end of the sentence; and ❷ a word followed by five values depicting the prosodic properties of the word.

All the prosodic properties are extracted by the decoder module of a pre-trained ProsodyLM (Qian et al., 2025), which is modified from a StyleTTS2 TTS synthesizer. It comes with an F0 contour extractor, an energy extractor, and a force-aligner (which can be used to extract duration information). All prosodic properties are quantized into 512 levels, forming new prosdy token vocabulary for size 512. A complete explanation of the tokenization scheme can be found in Qian et al. (2025).

## A.4. Inference

**System Prompt.** Since our method is designed for building an audio assistant, a system prompt will help the text-based LLM better understand its role and task. We therefore adopt the following system prompt:

```
You are a helpful assistant.  Always follow the user's instructions precisely and respond
with what is required.  Please read the instructions carefully before answering.  Your
responses must strictly align with the user's intent and constraints.
```

**Format Forcing.** During inference, we apply several *format-forcing logits processors* to ensure stable and well-structured generation:

- **Speaker F0-Median Processor.** To stabilize speech token generation, we enforce the model to emit the `<spk f0 med>` token immediately after `<speech>`. In addition, we constrain the first generated token to be non-speech.

- **Thinking Processor.** To ensure that the reasoning phase consists purely of text, we prohibit the model from generating any speech tokens while producing thinking content.

- **Deep Format-Forcing Processor.** After the model completes text generation, we enforce the output of speech tokens in a predefined format, which further stabilizes speech generation and ensures structural consistency.

- **Temperature Processor.** To encourage expressive audio generation, this processor applies different temperature values to the text and speech segments during decoding.

### A.5. Long Thinking

We empirically observed that even if a `<think>` token is provided as the context, SPEECHCOMBINE may not start the reasoning process, because it is unclear whether to perform reasoning, which is inherited from the text LLM's weight, or to perform text continuation behavior, which is strengthened during speech continuous pre-training. To disambiguate that, we will append *"Okay"* right after `<think>`, which is a common starting word of the long thinking process for QWEN models, to enhance the signal for long thinking.

During the thinking process, we ban the generation from the prosody token vocabulary, as we want the thinking process to be silent. Generation to verbalized thinking is easily achievable by removing the ban. Also, to prevent the model from aborting prematurely, we also ban the `</think>` token until the thinking reaches a pre-determined minimum length, which is set to 160 tokens in our experiments.

## B. Additional Experiment Details

### B.1. Text-Oriented Tasks

**Datasets.** We provide a brief introduction of the benchmark datasets on text-oriented tasks.

- `OpenbookQA (Voicebench)`: A speech-based multiple-choice science QA task evaluating knowledge-intensive reasoning under speech input, with performance measured by answer accuracy.

- `MMSU (VoiceBench)`: A speech-based multiple-choice understanding dataset covering diverse domains, evaluating a model's multimodal speech comprehension and reasoning ability, measured by answer accuracy.

- `Gsm8kEval (URO Bench)`: A speech-based math dataset evaluating multi-step numerical problem solving from spoken queries, measured using LLM-as-a-judge.

- `MLCpro-en (URO Bench)`: A speech logic and reasoning dataset assessing complex reasoning and instruction understanding, measured using LLM-as-a-judge.

- `TruthfulEval (URO Bench)`: A speech dataset measuring a model's ability to generate truthful and non-misleading answers under spoken prompts, measured using LLM-as-a-judge.

- `MLC (URO Bench)`: A speech-based benchmark evaluating general reasoning and comprehension, measured using LLM-as-a-judge.

**Task Prompts.** We find that many of the above benchmarks have very rigid evaluation schemes. If the output format deviates from their pre-defined templates, the answers will be considered incorrect, even though the output answers are still very reasonable. For example, on the math benchmarks, $\sqrt{3}$ is considered an incorrect answer whereas 1.732 is considered correct. While baseline models, which has been extensively trained to get accustomed to such hidden requirements, SPEECHCOMBINE is not familiar with any of such rules. Therefore, we need to introduce the following task prompt to regulate the output format of SPEECHCOMBINE.

- `OpenbookQA (Voicebench)`, and `MMSU (VoiceBench)`: *Please answer the following question. Please indicate which option (a, b, c, or d) is correct. Please put your answer in the format "The correct answer is: ___."*

- `MLCpro-en (URO Bench)`, `TruthfulEval (URO Bench)`, and `MLC (URO Bench)`: *Please answer the question very carefully and produce complete answers. If the answer contains any square root, convert it to decimal format.*

### B.2. Speech Understanding Tasks

**Datasets.** We provide a brief introduction of the benchmark datasets on speech understanding tasks.

- `UnderEmotion-en (URO Bench)`: A speech dataset that assess model's capability on emotion understanding task, measured using LLM-as-a-judge.

- EmphAssess Understanding: Since the original EmphAssess contains four different speakers saying the sentences with same emphasized words. To keep our inference cost tractable, we only keep speaker ex04's audio for evaluation. This dataset evaluates model's capability on emphasized words detection, measured by precision, Recall, and F1 Score.

**Task Prompts.** We adopt the following task prompts for the speech understanding tasks.

- UnderEmotion-en (URO Bench): *Please answer the question in the second person point of view. Please use an empathetic tone to answer the question.*

- EmphAssess Understanding: *Answer the question from the user. Put your answer in the format Emphasized word(s):*

For EmphAssess Understanding, the same prompt is applied to *all models*. For UnderEmotion-en, the task prompt provides more information than the format to elicit the prosody reasoning mode, as discussed in Section 4.3.

### B.3. Speech Generation Tasks

**Datasets.** We provide a brief introduction of the benchmark datasets on speech generation tasks.

- GenEmotion (URO Bench): A speech-based emotion generation benchmark evaluating a model's ability to generate speech with target emotional characteristics. The metric is defined as the target emotion probability predicted by an EMOTION2VEC classifier, weighted by $(1 - \text{WER})$ to account for content intelligibility.

- EmphAssess Generation: Since the original EmphAssess contains four different speakers saying the sentences with same emphasized words. We only keep speaker ex04's label to construct the EmphAssess Generation dataset. We then use COSYVOICE to synthesis text question *Please read the sentence: xxx with emphasis on the word: xxx* to audio. This datasets evluated model's emphasized word generation. We use EmphAssess's evaluation pipeline to compute the overall precision, recall, and F1 Score.

**Task Prompts.** For speech generation tasks, a strict word error rate is computed as part of the evaluation score. Any deviation from the requested content will be considered incorrect. For example, when the task requests uttering the sentence "How are you?". If the model answers *"Sure, here is the utterance..."*, which is still a reasonable answer, it will still be considered incorrect. Therefore, we adopt the following task prompts for the speech generation tasks to regulate the output format.

- GenEmotion (URO Bench): *In this task, you are asked to read a sentence in a given emotion. You MUST output only the exact sentence that should be spoken. The arousal level should be high.*

- EmphAssess Generation: *In this task, you are asked to read a sentence with emphasized words. You MUST output only the exact sentence that should be spoken. Please normalize number in your output.*

## C. Additional Experiment Results

### C.1. Ablation Study on $\lambda$ on more tasks

To further study the performance of SPEECHCOMBINE under different $\lambda$ values, we evaluate the performance on more tasks, shown in Tables 4 and 5. These results provide insights into the optimal selection of $\lambda$. Specifically –

- If the deployment goal focuses more on factual knowledge, reasoning, and other text-oriented skills, we recommend setting a $\lambda$ value between 0.6 and 0.85. A $\lambda$ below 0.6 would result in the model not recognizing the [speech] sections in its context; a $\lambda$ above 0.85 would result in the model starting to forget the text LLM's original knowledge.

- If the deployment goal focuses more on speech understanding and generation, we recommend setting a $\lambda$ value between 0.8 to 1. A $\lambda$ below 0.8 would result in compromised speech knowledge.

*Table 4.* Results on text-oriented tasks with different $\lambda$.

| $\lambda$ | OpenBookQA | MMSU | MLCPro |
|---|---|---|---|
| 0.60 | 89.89 | 77.78 | 90.10 |
| 0.65 | 88.35 | 76.41 | 91.57 |
| 0.70 | 88.13 | 76.64 | 89.37 |
| 0.75 | 87.03 | 76.35 | 93.04 |
| 0.80 | 86.81 | 74.26 | 90.84 |

*Table 5.* Results on speech understanding and generation tasks with different $\lambda$.

| | (a) **Speech Understanding** | | | | (b) **Speech Generation** | | | |
|---|---|---|---|---|---|---|---|---|
| | UnderEmo | Emph Detection | | | GenEmo | Emph Generation | | |
| $\lambda$ | *Acc.* ↑ | *Prec.* ↑ | *Recall* ↑ | *F1* ↑ | *Score* ↑ | *Prec.* ↑ | *Recall* ↑ | *F1* ↑ |
| 0.80 | 62.04 | 49.51 | 63.90 | 55.79 | 37.91 | 22.15 | 35.13 | 27.17 |
| 0.85 | 50.75 | 55.11 | 67.89 | 60.84 | 45.42 | 26.16 | 40.17 | 31.68 |
| 0.90 | 50.99 | 59.25 | 70.93 | 64.57 | 41.95 | 27.23 | 39.36 | 32.19 |
| 0.95 | 37.76 | 52.69 | 70.93 | 60.47 | 43.50 | 28.07 | 39.20 | 32.72 |
| 1.00 | 29.87 | 52.69 | 70.93 | 60.47 | 43.22 | 26.81 | 34.64 | 30.23 |

## C.2. Removing Format Forcing

To study the effect of removing format forcing, we performed an experiment on three representative tasks where we removed the format forcing. Note that the deep thinking section tends to degenerate into meaningless prosody tokens without format forcing. Hence, we disabled the thinking mode and evaluated the output stability of the rest of the output sequence. Table 6 shows the results.

As can be observed, removing format forcing produces almost no impact on the performance, except for the speech understanding task. This result suggests that while output instability is an issue for SpeechCombine, it is not devastating for most output segments. The major instability problem comes from the thinking section, which we will seek to address as a future direction.

*Table 6.* Effect of removing format forcing on representative tasks.

| | OpenBook | GenEmo | Emph Detect | | |
|---|---|---|---|---|---|
| **Model** | *Acc.* ↑ | *Score* ↑ | *Prec.* ↑ | *Recall* ↑ | *F1* ↑ |
| SPEECHCOMBINE (no think) | 64.83 | 36.37 | 42.90 | 69.78 | 53.14 |
| SPEECHCOMBINE (no format forcing) | 65.71 | 36.36 | 20.38 | 43.34 | 27.73 |

## C.3. Generalization Across Other Model Families

To study SPEECHCOMBINE's generalization across non-QWEN model families, we performed experiments on (1) OLMO-3-7B-THINK, (2) LLAMA-3.1-8B. The results on representative tasks in the three task types are shown in Table 7.

As can be observed, the generalization to other model families is successful, bounded by the capability constraint of the corresponding text model (the text models of OLMO and LLAMA are significantly weaker than QWEN. Hence the corresponding SpeechCombine model underperforms QWEN as well). The only exception is the GenEmo performance of OLMO, for which we observed a convergence issue on the prosody token loss during the continuous pre-training. Nevertheless, these results confirm the generality of the SPEECHCOMBINE framework.

*Table 7.* Generalization of SPEECHCOMBINE to different text model families.

| Model | OpenBook | GenEmo | Emph Detect | | |
| --- | --- | --- | --- | --- | --- |
| | *Acc.* ↑ | *Score* ↑ | *Prec.* ↑ | *Recall* ↑ | *F1* ↑ |
| SPEECHCOMBINE-QWEN | 86.59 | 45.42 | 55.11 | 67.89 | 60.84 |
| SPEECHCOMBINE-OLMO | 72.52 | 21.35 | 28.18 | 37.88 | 32.32 |
| SPEECHCOMBINE-LLAMA | 62.85 | 32.34 | 22.79 | 59.92 | 33.02 |

