# OpenReview forum: "Unlocking Speech–Text Compositional Powers: Instruction-Following Speech Language Models without Instruction Tuning"
_ICML.cc/2026/Conference — ICML 2026 regular_

### Official Review · Reviewer_RhVK · 2026-02-25

**Soundness:** 3
**Presentation:** 2
**Significance:** 3
**Originality:** 4
**Overall Recommendation:** 4
**Confidence:** 4

**Summary:**

The paper presents a method to achieve speech-instruction following abilities in speech LMs without using speech instruction data. This is demonstrated across 3 types of speech instruction tasks: text instructions (synthesised as speech), speech understanding and speech synthesis. The proposed approach achieves this by weighted model averaging between the base text LM, its (text) instruction following version, and a speech-text pre-training version starting from the pre-trained text LLM. This model averaging is joint with a specific representation for the speech-text data, which includes the text and caption, as well as inference prompting to elicit the wanted behaviour.

**Compliance With Llm Reviewing Policy:**

Affirmed.

**Final Justification:**

The authors rebuttal increased the soundness of the evaluation explaining where much of the performance gains come from. While it shows that some of the gains are related to the use of an external ASR, it also demonstrates definite generalisation abilities. This idea can be of interest to the community in understanding and training instruction tuning SpeechLLMs.

In my opinion the writing of the paper still needs improving, which might be beyond the scope of a rebuttal.

**Key Questions For Authors:**

**Soundness related questions**
1) Table 1 - I think it would add to have a cascaded "ASR -> TextLLM (-> Optionally) TTS" baseline since for the text based tasks this should achieve very good scores. Since the proposed approach also uses an external ASR model at inference this is a relevant baseline to assess whether the model merging is actually helping, or rather using a strong ASR model with a strong LLM leads to good performance.
2) The proposed model, potentially internalised the captioning task (from pre-training) and therefore could be implicitly learning a cascaded pipeline of audio captioning (irrespective of the specific question), and then reasoning in text. Figure 3 (c) somewhat suggests this as the model described the speaking rate and arousal and not directly the stressed words it was asked for. Is that observed in other cases? Could the authors quantify how well the model is at the task of captioning? Perhaps considering a baseline of performing the tasks at hand based on ASR + captions (even those produced by the merged model) to see whether this information is sufficient to the text LLM. It is cool that the model can internalise and merge these abilities simply by model merging, but this could help clarify the mechanism by which this transfer happens.
3) Tables 1 & 2 - why is GPT-4o on a separate line and not part of the "bolds". It is not a "top line", rather just another baseline?
4) Adding some other "families" of SpeechLMs to evals might be useful, e.g Moshi [1].
5) Might be interesting to evaluate stress detection and understanding on real datasets and not only synthetic EmphAsses, e.g. [2].

**Writing suggestions**
1) In my opinion, the paper is quite slim in citations for backing broad claims such as in the intro. For instance L43-47 (right column) - *"The training datasets typically consist of discrete speech tokens derived from both real speech and synthetic speech generated from text datasets originally used for text LLM training."* - this should be backed by some citations. As far as I know this is somewhat of a simplification as some pre-training or instruction datasets are also specifically synthesised for speech and not based on text datasets, (e.g [2], and others). Another example - *"Specifically, an SLM is first pre-trained on a large speech corpus using a next-token prediction objective, and then fine-tuned on datasets containing various speech instructions using supervised fine-tuning (SFT) and/or reinforcement learning (RL)"* - this seems to ignore a key training paradigm often noted Speech-Aware LMs which do not do any pre-training on speech, instead use an instruction tuned text LM, and only conduct speech instruction tuning, sometimes only training some modality adapter.
2) The paper seems to ignore many existing SpeechLMs, e.g L52 (Right column) *"speech utterance typically lasts around one second and expands to roughly 60–200 speech tokens"* - while there are definitely some tokenisers for which this is true there are many speech tokens used in SpeechLMs for which this is up to 25 tok/sec.
3) L59 left column - *"leading to notable compromises in both knowledge acquisition and instruction-following capabilities."* - this claim is somewhat unsupported, there could be many reasons for less effective instruction tuning capabilities, it is not clear that the presented hypothesis is the main or only reason.

[1] Défossez, Alexandre, et al. "Moshi: a speech-text foundation model for real-time dialogue." arXiv preprint arXiv:2410.00037 (2024).

[2] Yosha, Iddo et al. "Stresstest: Can your speech lm handle the stress?." arXiv preprint arXiv:2505.22765 (2025).

[3] Arora, Siddhant, et al. "On the landscape of spoken language models: A comprehensive survey." arXiv preprint arXiv:2504.08528 (2025).

**Limitations:**

yes

**Strengths And Weaknesses:**

**Strengths**
- *Originality* - This paper introduces an exciting new approach for optimising instruction following speech langauge models, which is under explored.
- *Significance* - The approach is relatively simple to train, and has the potential to address important existing challenges in SpeechLMs such as text knowledge forgetting.

**Weaknesses**

My main concerns about the paper relate to the *soundness*, i.e. to what extent the results reported back the key claim of instruction following abilities. *I will happily increase my score if these are addressed*, see below and "Key questions" for specific details.
- It is unclear to me to what extent the model might be able to rely solely on text reasoning abilities together with the use of external transcriptions to achieve the described abilities. For the "text-oriented" instructions this is trivially true, but even for the speech understanding instructions. E.g the stressed word could be the only likely option in a given sentence or the sentiment could be implied by the sentence in some test cases. It would greatly strengthen the work to compare with an ASR + LLM baseline using the same ASR as used in your pipeline and the same reasoning LLM. It could also help to consider "adversarial" datasets where the text information is not beneficial for the task or even contrarian, e.g stress detection & reasoning in [1].
- It is not fully clear to me whether the speech tasks truly need "instruction following" abilities as the proposed pre-training dataset implicitly trains models for synthesis given a text and description or for audio captioning given a speech segment and the transcription. Might be worth measuring and comparing to the abilities of the pre-trained speech model itself by advanced prompting, e.g emotion recognition or stress detection might be described a few shot prompt and solved well via in-context learning for the speech only model without composition, due to the suggested training paradigm.

[1] Yosha, Iddo et al. "Stresstest: Can your speech lm handle the stress?." arXiv preprint arXiv:2505.22765 (2025).

---

> ### Author Rebuttal · Authors · 2026-03-31
>
> We would like to thank reviewer RhVK for the constructive feedback on sounded and writing. We address the main concerns below.
>
> ### **W1 & Soundness Q1 & Q5: Cascaded ASR -> TextLLM Baseline + StressTest Benchmark**
>
> Following your suggestion, we add the ASR -> TextLLM as a new baseline and evaluate it on 2 text-oriented tasks and 2 speech understanding tasks, including the new StressTest benchmark. The results are as follows
>
> |  | openbookQA | MMSU | EmphAssess (Understand) | StressTest |
> |------|------|------|------|------|
> | SpeechCombine       | 86.59   | 73.38   | 55.11/67.89/60.84   | 20.02/64.20/30.52 |
> | ASR -> Text LLM | 83.29 | 73.22 | 15.79/26.94/19.91 | 18.72/61.99/28.76 |
>
> The results, as expected, show that the ASR->Text LLM can achieve superior performance on text-oriented tasks, but performs very poorly on speech understanding tasks. This is because it tends to **infer the stress from text**, rather than from the audio. Full results will be added to paper.
>
> ### **W2: Removing Instruction Vector**
>
> Following your suggestion, we add another method where we remove the instruction-tuning vector, but instead insert in-context learning examples to teach it QA capabilities. We only evaluate on three representative tasks due to time limitations and the tremendous number of new experiments. The results are as follows
> |  | openbookQA | Emph Detection | GenEmotion |
> |------|------|------|------|
> | SpeechCombine (no think)      | 64.83   | 42.9/69.78/53.14  | 36.37   |
> | Replace instruct vector with in-context examples | 71.68   | 47.78/34.96/40.38 | 15.34  |
>
> The results show that in-context examples can achieve better performance on text-oriented task, but worse for speech understanding & generation, which suggests it cannot completely replace instruction vector. Moreover, this paradigm **cannot incorporate deep thinking**, and requires **task-dependent** in-context learning, which is far less flexible and powerful than the direct instruction-following and thinking capabilities as in SpeechCombine. Full results will be added to paper.
>
> ### **Soundness Q2: Internalizing Captioning**
>
> Following your suggestion, to test whether SpeechCombine effectively internalizes speech captioning, we add another method where we use SpeechCombine to produce a caption of the input speech, and then let the QWEN3-Instruct to answer the EmphAssess questions. The performance is 25.31/44.49/32.27. Although worse than SpeechCombine (F1: 60.84), it is far better than ASR->Text LLM (F1: 19.91). Considering speech captioning is trained to not always include stressed words, this result shows that the speech captions produced by SpeechCombine are very meaningful.
>
> ### **Soundness Q3: Why is GPT-4o Separated from Bolding?**
>
> We excluded GPT-4o from direct comparison because it is much larger in size. All the other methods are around 8B in size, but GPT-4o is estimated to be much larger than 8B. Knowledge and instruction-following capabilities tend to grow with model size, so that gives GPT-4o unfair advantages.
>
> ### **Soundness Q4: More Speech LM baselines**
>
> We initially did not include Moshi because it is rather old and underperforms the other baselines we included. However, following the reviewers' suggestion, we added two newer and more powerful speech LMs as new baselines. Please refer to **Response to Reviewer RzSR, W2** for the results.
>
> ### **Writing Q1: Insufficient Backing for Broad Claims**
>
> We genuinely appreciate your suggestion on this. Specifically --
>
> * *"The training datasets ... both real speech and synthetic speech ... for text LLM training."*
>
>     We explicitly mentioned **"real speech"** and **"synthetic speech"** in this sentence for broadest coverage. We will add [[1](https://aclanthology.org/2025.acl-long.681.pdf), [2](https://arxiv.org/abs/2505.22765)] after "real speech", and [[3](https://arxiv.org/abs/2409.06666), [4](https://arxiv.org/abs/2408.16725)] after "synthetic speech".
> * *"Specifically, an SLM is first pre-trained ... reinforcement learning (RL)"*
>
>     We will change it to "Specifically, **in a mainstream training paradigm**, a SLM is first pre-trained ... (RL) **[[5](https://arxiv.org/pdf/2507.08128), [6](https://arxiv.org/pdf/2504.18425), [7](https://arxiv.org/pdf/2512.20156)]. Some others only perform speech instruction-tuning [[8](https://arxiv.org/pdf/2305.11000)].**"
>
> ### **Writing Q2: Token Rate for Speech Tokenizer**
>
> One caveat is that while many speech tokenizers claim that their code rate is 25Hz, **each speech code typically consists of 8 tokens**, so the actual token rate would be 25 * 8 = 200 (Hz) [[9](https://arxiv.org/pdf/2412.02612), [10](https://arxiv.org/pdf/2506.23325v1)]. It was not until very recently that a genuine 25-Hz tokenizer was proposed [[11](https://arxiv.org/pdf/2502.04465)]. We will modify our claim accordingly.
>
> ### **Writing Q3: L59 Left**
>
> We will soften this claim and conduct an actual calculation of equivalent text tokens to support this claim.

---

> > ### Author Rebuttal · Reviewer_RhVK · 2026-04-02
> >
> > I thank the authors for their detailed response.
> >
> > I think the cascaded ASR->LLM pipeline results are very important to show which abilities are simply learned due to the use of ASR and strong LLM vs tasks which can only be inferred from speech. The latter are those which truly show the instruction abilities in speech. I hope the authors will add these results as well as the in context learning results, captioning results and additional baselines to the final version.
> >
> > These have addressed most of my soundness related claims. While they disclose that for some of the tasks indeed the ASR itself or the internalisation of captioning from the pre-training objective leads to the performance, it further highlights the value of the model merging.
> >
> > Regarding writing - the point for correction was that synthetic data is not necessarily used for text LLM training (it is not a text dataset + TTS). Anyway, while I appreciate the comments writing is inherently harder to fully discuss during a rebuttal.
> >
> > I have increased my soundness score and my overall score accordingly.

---

### Official Review · Reviewer_RzSR · 2026-03-10

**Soundness:** 2
**Presentation:** 3
**Significance:** 2
**Originality:** 2
**Overall Recommendation:** 3
**Confidence:** 5

**Summary:**

This paper proposes SPEECHCOMBINE, a simple training approach for speech language models that avoids explicit speech instruction tuning. The method computes two parameter directions: one from speech pretraining and one from the instruction-tuned version of a text LLM. These two directions are then combined in weight space to produce a speech-capable model that is expected to inherit instruction-following ability from the text LLM. The authors train the model with a single round of speech pretraining on about 30k hours of data and evaluate it on several QA, reasoning, speech understanding, and speech generation tasks.

**Compliance With Llm Reviewing Policy:**

Affirmed.

**Key Questions For Authors:**

1. How does SPEECHCOMBINE compare with recent speech instruction-following approaches such as InSerter or other interleaved speech-text pretraining methods?

2. Would the method still work if the text transcription were removed? In other words, does the model truly reason over speech signals, or mainly rely on the text input?

3. How much improvement actually comes from the instruction vector vs. the speech vector? A clearer ablation isolating each component would help.

4. How does the method compare against stronger recent speech-LLM baselines such as Qwen2.5-Omni or newer audio-LLM systems?

5. Is the weight merging approach consistently better than standard fine-tuning or joint pretraining under similar compute and data settings?

**Limitations:**

Same as the weaknesses I mentioned.

**Strengths And Weaknesses:**

Strengths:
1. The paper explores a simple and lightweight alternative to the common multi-stage SLM training pipeline.
2. The approach is easy to implement and computationally cheap compared with large-scale speech instruction tuning.
3. The experiments cover multiple task types, including both text-oriented tasks and speech-related tasks.
4. The idea of transferring instruction-following ability from text LLMs to speech models through weight composition is interesting.

Weaknesses:

1. Limited novelty. The method mainly applies a straightforward form of weight arithmetic / model merging. The paper does not introduce new learning mechanisms or theoretical insights beyond existing task arithmetic approaches.

2. Evaluation is incomplete. Very limited baselines shown in Table 1&2. Moreover, the comparisons include several relatively old systems (e.g., OSUM-EChat, GLM-4-Voice) and miss many recent strong speech-LLM models such as Qwen2.5-Omni, Flamingo3Audio, or StepAudio-R1. As a result, it is hard to judge the real competitiveness of the approach.

3. Positioning with prior work is weak. The paper does not provide a clear comparison with recent speech instruction-following related works. The differences in capability transfer are not analyzed.

4. Speech reasoning is unclear. The model relies on external ASR to convert speech into text, which makes it unclear whether the system actually performs reasoning over speech signals or mostly operates on text.

5. Limited ablation and analysis. Important questions about the merging strategy are not thoroughly explored (e.g., the role of each direction vector, comparison with standard fine-tuning, or sensitivity to merging weights).

---

> ### Author Rebuttal · Authors · 2026-03-31
>
> We would like to thank reviewer RzSR for the detailed comment and suggestion. We address the main concerns below.
>
> ### **W1: Novelty**
>
> We would like to clarify that SpeechCombine is far more than just a naive weight-combined model, but rather a complex system with many intricate designs and unexpected findings. Just to name a few here
>
> - **Pre-training Designs**. The key to SpeechCombine's success relies on a speech pre-training scheme, which is unconventional in many ways:
>
>     1. *Unconventional randomized `[cap]` both before and after `[speech]`.* Conventional speech-text interleaved pre-training does not involve such a design, but this turns out to be crucial for infusing speech generation and understanding knowledge.
>     Please refer to **Response to Reviewer TTCr, W1, Table row 2** for our **new experiments on removing `[cap]`**, where speech performance worsens significantly.
>     2. *[text] as speech-text combine anchors.* In conventional text-speech interleave pre-training, `[text]` serves only one purpose: guide the learning of `[speech]` into the same semantic space. In SpeechCombine, `[text]` is mandatory because it serves as an anchor of the weight-combined model (This is related to your concerns on ASR; please see W3 for further explanation).
>     Please refer **Response to Reviewer TTCr, W1, Table row 3** for our **new experiments on removing `[text]`**.
>     4. *Unconventional speech tokenization*. Unlike conventional waveform tokenization schemes, we adopt the prosody tokenization scheme to minimize redundancy over the mandatory `[text]` section.
> - **Deep Combination.** Existing weght combination mostly focused on combining parallel skills. We are first to explore **skill-knowledge combination across speech & text modalities**. The deep combination capabities are suprsing: Instruction-following and deep thinking can work on speech even without trained on speech queries. See Figure 3 for one such surprising finding.
>
> ### **W2 & Q1 & Q4: Newer Baselines**
>
> We add the following new baselines to our comparison: Qwen2.5-Omni, Audio Flamingo 3, and Inserter (results copied from paper for 2 tasks only because it is not open-weight). We did not include Step-Audio-R1 for now because it is a much larger model (32B). The results are as follows.
> *Text-Oriented*
> |  | openbookQA | MMSU | MLCPro |
> |------|------|------|------|
> | SpeechCombine       | 86.59   | 73.38   | 89.01   |
> | Qwen2.5-Omni        | 81.1   | 61.32   | 65.2   |
> | Audio Flamingo 3    | 58.68   | 42.19   | 61.9   |
> | Inserter    | 77.14   | 59.27   | -   |
>
> *Speech Undertanding and Generation*
> | | Emph Detection | Emph Generation | GenEmotion| UnderEmo|
> |------|------|------|------|------|
> | SpeechCombine       | 55.11/67.89/60.84   | 26.16/40.17/31.68    |45.42|52.70|
> | Qwen2.5-Omni        | 3.75/26.26/6.56   | 18.82/34.03/24.24    |27.18|33.91|
> | Audio Flamingo 3$^*$    | 17.09/33.32/22.59   | -    |- |24.28|
>
> $^*$ Speech generation unavailable because the speech decoder code is not released
>
> SpeechCombine remains competitive when these new baselines are added.
>
> ### **W3: Positioning with Prior Work**
>
> We will add all the new baselines to the prior work section and revise our discussion on how SpeechCombine differs from these works in obtaining the new capabilities.
>
> A brief gist: Existing SLM typically acquire speech instruction-following via Pre-training, Adaptor Training, and Fine-Tuning. By weight combination and careful pre-training designs, we avoid the redundant training steps and data to remind the model of knowledge and instruction following skills it already knows. Hence, a single training pass on 30k hours of speech suffices.
>
> ### **W4 & Q2: Reasoning over speech or text?**
>
> As discussed in W1, removing `[text]` will significantly impact performance, not because SpeechCombine only reasons over text, but because `[text]` serves as an anchor to fuse intruct-following skills and speech knowledge.
>
> To address your concern about whether SpeechCombine reasons over speech, we perform another experiment where we remove `[speech]`. The results on EmphAssess (understand) are 14.25/32.57/19.83, which confirms that removing `[speech]` in the question significantly impacts speech understanding performance, confirming that reasoning over speech is essential for SpeechCombine's success.
>
> ### **W5 & Q3 & Q5: More Ablation Studies**
>
> * Please refer to **Response to Reviewer rv6E, W1** for **new comparison with direct fine-tuning and other baselines**.
> * Please refer to **Response to Reviewer rv6E, W2** for **additional sensitivity analysis of $\lambda$**.
> * Please refer to **Response to Reviewer RhVK, W2** for **removing the instruction vector**.
> * The following table shows the results when removing the speech vector:
>
> | openbookQA | Emph Detection | GenEmotion |
> |------|------|------|
> | 84.83   | Fail (cannot understand speech)  | Fail (cannot generate speech) |
>
> which shows the speech vector is essential for speech understanding tasks.

---

> > ### Author Rebuttal · Reviewer_RzSR · 2026-04-06
> >
> > Thank the authors for the rebuttal. After reviewing, I'd like to keep my original score.

---

> > > ### Author Response · Authors · 2026-04-06
> > >
> > > Dear reviewer RzSR,
> > >
> > > Thank you for your constructive comment and for taking the time to review our paper. During the rebuttal acknowledgement phrase, you indicate that all of your concerns have been properly addressed and do not raise any additional concerns. As you mentioned, the proposed method is both **interesting** and **computationally cheap**. Given that there are no new concerns have been raised, indicating the overall strengths of the paper outweigh its weaknesses, we would be grateful if you could reconsider the score to better reflect the positive evaluation.
> > >
> > > Sincerely,
> > >
> > > Authors

---

### Official Review · Reviewer_TTCr · 2026-03-12

**Soundness:** 2
**Presentation:** 3
**Significance:** 2
**Originality:** 3
**Overall Recommendation:** 3
**Confidence:** 4

**Summary:**

The paper introduces SpeechCombine, a speech language model that can follow instructions without any instruction‑tuning, using only a single round of speech pre‑training on 30k hours of data. The method begins with a text LLM that the authors pre‐train on speech to obtain a speech‑adapted model.  They then add two weight‑difference vectors consisting of the the speech‑adaptation direction and the instruction‑following direction derived from the text LLM. Combining these two parameter directions yields a model that simultaneously acquires speech‑processing abilities and retains the instruction‑following skills of the original text model. Despite its simplicity and modest data requirements, SpeechCombine delivers competitive or state‑of‑the‑art performance across multiple benchmarks while using less than 1% of the data consumed by existing speech LLMs.

**Compliance With Llm Reviewing Policy:**

Affirmed.

**Final Justification:**

See rebuttal comments

**Key Questions For Authors:**

- Does this model merging scheme work on other models than Qwen3?

- To ensure that we still retain most of the capabilities of Qwen3, can you provide us with results where lambda equals 0 in Figure 4?

**Limitations:**

yes

**Strengths And Weaknesses:**

Strengths:
+ Proposes an interesting model merging approach that allows for both preserving existing knowledge capabilities while also acquiring new speech specific skills.
+ The approach drastically reduces the amount of speech training data required for instruction-tuning SLM.

Weaknesses:
- A lack of validation on the pretraining data structure shown in section 3.3.  For instance, how does the approach perform if you remove [cap] or [text]?
- The format forcing methodology is ad-hoc and indicates instability.  Could you provide an ablation of results with and without format forcing?
- It is unclear from the results how the weights of SpeechCombine differ from the baseline Qwen3.  Could the authors quantify or visualize how the weights change between the baseline, the instruction baseline, the speech baseline and the combined SpeechCombine?
- The weight merging method is only validated on Qwen3.  It is unclear if the approach could generalize to other language models.

---

> ### Author Rebuttal · Authors · 2026-03-31
>
> We would like to thank Reviewer TTCr for the constructive comments and suggestions. We address the main concerns below.
>
> ### **W1: Ablation studies on data structure**
>
> Following your suggestions, we conduct additional ablation studies on (1) removing `[cap]`, (2) removing `[text]`. The results are as follows
>
> *Text-Oriented*
> |  | openbookQA | MMSU | MLCPro |
> |------|------|------|------|
> | SpeechCombine       | 86.59   | 73.38   | 89.01   |
> | w/o `[cap]`  | 84.83   | 73.19   | 81.31   |
> | w/o `[text]`        | 83.95   | 71.56   | 89.01   |
>
> *Speech Understanding and Generation*
> |  | Emph Detection | Emph Generation | GenEmotion| UnderEmo|
> |------|------|------|------|------|
> | SpeechCombine       | 55.11/67.89/60.84   | 26.16/40.17/31.68    |45.42|52.70|
> | w/o `[cap]`  | 3.08/0.21/0.39   | 18.81/27.81/22.44    |27.18|59.22|
> | w/o `[text]`        | 4.76/0.21/0.4   | 25.86/35.98/30.09    |35.69|56.49|
>
> Here are the key takeaways:
>
> - When the `[cap]` is removed, the speech generation and understanding performance degrades significantly, which confirms that `[cap]` introduces the knowledge of speech generation and understanding.
> - When `[text]` is removed, the speech generation and understanding performance is also significantly impacted, which is consistent with our conclusion that `[text]` serves as an anchor for fusing speech knowledge. The impact is less severe than removing `[cap]` because the ability to understand text is already inherent in the text LLM models.
>
> ### **W2: Removing Format Forcing**
>
> Following your suggestion, we added an experiment on three representative tasks where we removed the format forcing. Note that the deep thinking section tends to degenerate into meaningless prosody tokens without format forcing. Hence, we disabled the thinking mode and evaluated the output stability of the rest of the output sequence. The results are as follows.
>
> | Model | openbookQA | GenEmotion | Emphassess (Understand) |
> |------|------|------|------|
> | SpeechCombine(no think)   |  64.83   | 36.37   | 42.9/69.78/53.14  |
> | SpeechCombine (no format forcing)| 65.71   |  36.36  | 20.38/43.34/27.73   |
>
> As can be observed, removing format forcing produces almost no impact on the performance, except for the speech understanding task. This result suggests that while output instability is an issue for SpeechCombine, it is not devastating for most output segments. The major instability problem comes from the thinking section, which we will seek to address as a future direction.
>
>
> ### **W3: Visualization of Weight Changes**
>
> We conducted a visualization of weight changes, which shows the interesting complementarity of the two directions. We cannot post the figures here, but we will add them to our paper.
>
> ### **W4 & Q1: Non-QWEN Model Families**
>
> To verify whether SpeechCombine also works for other model families/sizes, we add new experiments on (1) [Olmo-3-7B-Think](https://huggingface.co/allenai/Olmo-3-7B-Think), (2) [Llama-3.1-8B](https://huggingface.co/meta-llama/Llama-3.1-8B).  Unfortunately, we are unable to finish these experiments by the 1st-round rebuttal deadline due to the extremely large volume of experiments. We will report these results in our next round of response.
>
> ### **Q2: Setting $\lambda$ to 0**
>
> We would like to clarify that setting $\lambda = 0$ alone will **not** restore the text LLM's performance even on text-oriented tasks, but would result in complete failure. This is because the input to the model includes not only `[text]` but also `[speech]` sections. Setting $\lambda = 0$ would make the model completely unable to recognize the `[speech]` sections and produce degenerate output.
>
> On the other hand, we confirmed that setting $\lambda = 0$ **and** removing all the `[speech]` sections in the input questions can restore the text LLM's performance on text-oriented tasks.

---

> > ### Author Rebuttal · Reviewer_TTCr · 2026-04-03
> >
> > Thank the authors for the rebuttal. After reviewing, I'd like to keep my original score.

---

> > > ### Author Response · Authors · 2026-04-06
> > >
> > > As promised, we would like to provide our new results on **non-QWEN model families**. Specifically, we add new experiments on (1) Olmo-3-7B-Think, (2) Llama-3.1-8B. The results on representative tasks in the three task types are shown in the following table
> > >
> > > | | OpenbookQA | GenEmo | EmphAsses Understand |
> > > |----|----|----|----|
> > > | SpeechCombine-QWEN |86.59 |45.42 |55.11/67.89/60.84|
> > > | SpeechCombine-Olmo |72.52 | 21.35| 28.18/37.88/32.32|
> > > | SpeechCombine-Llama |62.85 |32.34 | 22.79/59.92/33.02|
> > >
> > > As can be observed, the generalization to other model families is successful, bounded by the capability constraint of the corresponding text model (the text models of Olmo and Llama are significantly weaker than QWEN. Hence the corresponding SpeechCombine model underperforms QWEN as well). The only exception is the GenEmo performance of Olmo, for which we observed a convergence issue on the prosody token loss during the continuous pre-training. Nevertheless, these results confirm the generality of the SpeechCombine framework.
> > >
> > > ---
> > >
> > > Meanwhile, we noticed that your reviews on our paper are **generally positive**, despite the unchanged score. Specifically, based on your review and responses -
> > >
> > > * SpeechCombine is interesting and data efficient;
> > > * All concerns have been addressed (otherwise (b) or \(c) would have been chosen in the rebuttal acknowledgement)
> > > * No new concerns emerge (otherwise they would have been specified in the `Reasons` section)
> > >
> > > The only gap in the original rebuttal was the missing non-QWEN model families' results (which we conjecture is the main reason for holding your score). We hope that the new results provided above could already address this gap.
> > >
> > > However, in case you have other unspecified reservations, we would like to bring to your attention that, in addition to the merits that you already acknowledged, our new experiments also demonstrate some **other encouraging capabilities of SpeechCombine**. For example -
> > >
> > > * SpeechCombine outperforms many **new baselines**, including Qwen2.5-Omni, Audio Flamingo 3, and Inserter (See **response to RzSR, W2**)
> > > * SpeechCombine significantly outperforms many **direct pre-training and fine-tuning baselines** under similar data regime (See **response to rv6E, W1**)
> > >
> > > We hope these new merits can further add to your overall positive evaluation of the paper and justify a score that matches the positive evaluation. Thank you for your time!

---

### Official Review · Reviewer_rv6E · 2026-03-15

**Soundness:** 2
**Presentation:** 3
**Significance:** 2
**Originality:** 2
**Overall Recommendation:** 3
**Confidence:** 4

**Summary:**

This paper proposes SPEECHCOMBINE, a speech language model built by combining two weight-space directions: a speech adaptation vector obtained by continual pre-training a base text LLM on about 30k hours of speech data, and an instruction-following vector computed from the difference between the base and instruct versions of the same text LLM.

**Compliance With Llm Reviewing Policy:**

Affirmed.

**Final Justification:**

Thanks for the follow-up experiments. I would like to raise my score from 3 to 3.5.

**Key Questions For Authors:**

1. How does SPEECHCOMBINE compare against direct alternatives under the same data budget, especially: starting speech pre-training from the instruct model instead of the base model, simple checkpoint averaging, and modest speech instruction tuning after pre-training? These controls would strongly affect my assessment of the method’s necessity.
2. The paper validates the proposed linear composition only for a single instruction-following direction, namely base-to-instruct. It remains unclear whether the same hypothesis extends to other useful text-side deltas, such as instruct-to-RLHF/preference-optimized models or other specialized capability vectors. Testing these alternatives would help establish whether the method reflects a broader principle of cross-modal steering or a narrower checkpoint-specific effect.

**Limitations:**

yes

**Strengths And Weaknesses:**

Strengths:
1. The paper presents a simple and interesting training alternative to the standard pre-train plus speech instruction-tune pipeline, and the weight-space composition idea is conceptually clean.
2. The empirical coverage is broad, spanning text-oriented tasks, speech understanding, speech generation, qualitative reasoning traces, and a merge-weight ablation.

Weaknesses:
1. The core causal claim is not fully isolated: the paper lacks stronger direct baselines against simpler alternatives such as continuing from the instruct model, merging other checkpoints, or doing a small amount of speech instruction tuning with the same data budget.
2. The merge coefficient introduces a clear trade-off between retaining text-side reasoning ability and acquiring speech-specific competence. While Figure 4 shows a broad sweet spot, the paper does not provide a principled strategy for selecting this coefficient under different deployment goals, nor does it quantify how sensitive the final conclusions are to this choice.
3. The method is entirely training-free at the speech-instruction stage, yet the paper does not compare against training-based follow-up variants. It is therefore unclear whether the proposed merge is best understood as a full alternative to instruction tuning or as a strong initialization for further adaptation.

---

> ### Author Rebuttal · Authors · 2026-03-31
>
> We would like to thank reviewer rv6E for your constructive comments and questions. We address the main concerns below:
>
> ### **W1 & Q1: More Direct Baselines**
>
> Thank you for your suggestion. We followed the advice and added two more baselines for comparison:
>
> 1. **Continuous Pre-train**. Continuous pre-training on the `QWEN3-instruct` model (instead of base), using the same 30k hours of speech data.
> 2. **Continuous Pre-train+SFT**. Further fine-tune SpeechCombine(Instruct) on around 10,000 hours of speech instruction data from SIFT-50M[[1](https://arxiv.org/abs/2504.09081)], InstructS2S-200K[[2](https://arxiv.org/abs/2409.06666)], VoiceAssistant-400K[[3](https://arxiv.org/abs/2408.16725)], and do SFT on them.
>
> Due to the extremely limited time frame, we only perform the comparisons on openbookQA, GenEmotion, and Emphassess (Understand). We will add the results for all the datasets to the paper when they are ready. The results are shown below. The original SpeechCombine results are copied here for comparison.
>
>
> | Model | openbookQA | GenEmotion | Emphassess (Understand) |
> |------|------|------|------|
> | SpeechCombine   | 86.59   | 45.42   |  55.11/67.89/60.84  |
> | Continuous Pre-train   | 78.46   |  0.72  |   1.53/6.51/2.48$^*$ |
> | Continuous Pre-train + SFT   | 80.21   |  23.89  | 2.12/3.42/2.61  |
>
> $^*$ Output forced to one sentence because it does not know how to stop.
>
> The results suggest that the original SpeechCombine significantly outperforms these direct training baselines.
>
>
> ### **W2: Merge Coefficient ($\lambda$) Selection**
>
> Thank you for your suggestion.
>
> First, regarding the **general $\lambda$ coeddifficent selection strategy**, we will add a section on the selection strategy of $\lambda$ in our paper. The selection principles are as follows --
>
> - If the deployment goal focuses more on *factual knowledge, reasoning, and other text-oriented skills*, we recommend setting a $\lambda$ value between 0.6 and 0.85. A $\lambda$ below 0.6 would result in the model not recognizing the `[speech]` sections in its context; a $\lambda$ above 0.85 would result in the model starting to forget the text LLM's original knowledge.
> - If the deployment goal focuses more on *speech understanding and generation*, we recommend setting a $\lambda$ value between 0.8 to 1. A $\lambda$ below 0.8 would result in compromised speech knowledge.
>
> Second, to **empirically validate the above selection strategy**, we add another ablation study where we vary $\lambda$ and evaluate the performance on **all datasets** (instead of only 3 as in Figure 4). The results are shown below:
>
> *Text-Oriented*
> | $\lambda$ | openbookQA | MMSU | MLCPro |
> |------|------|------|------|
> | 0.60   | 89.89   | 77.78   | 90.10   |
> | 0.65   | 88.35   | 76.41   | 91.57   |
> | 0.70   | 88.13   | 76.64   | 89.37   |
> | 0.75   | 87.03   | 76.35   | 93.04   |
> | 0.80   | 86.81   | 74.26   | 90.84   |
>
> *Speech Understanding and Generation*
> | $\lambda$ | Emph Detection | Emph Generation | GenEmotion| UnderEmo|
> |------|------|------|------|------|
> | 0.80   | 49.51/63.9/55.79   | 22.15/35.13/27.17    |37.91|62.04|
> | 0.85   | 55.11/67.89/60.84   | 26.16/40.17/31.68    |45.42| 50.75|
> | 0.90   | 59.25/70.93/64.57   | 27.23/39.36/32.19    | 41.95| 50.99|
> | 0.95   | 52.69/70.93/60.47   | 28.07/39.20/32.72   | 43.5|37.76|
> | 1.00   | 52.69/70.93/60.47   | 26.81/34.64/30.23    |43.22|29.87|
>
>
> The results show that the proposed $\lambda$ range works well under different tasks.
>
>
> ### **W3: Alternative vs. Initialization**
>
> Given its strong performance, SpeechCombine is sufficient to be adopted as is. However, to verify whether it can also be applied as an initialization for further fine-tuning, we add an experiment where we fine-tune the original SpeechCombine on the speech instruct-tuning dataset introduced in W1. The results are as follows
> | Model | openbookQA | GenEmotion | Emphassess (Understand) |
> |------|------|------|------|
> | SpeechCombine + SFT  | 79.12   | 22.56   | 4.55/0.56/1.01  |
>
> After SFT, the performance becomes worse. This is likely due to the insufficient scale of fine-tuning. These results show that SpeechCombine is good to be used as is. Serving as an SFT initialization is still possible, but only when SFT is extended to very large scales.
>
>
> ### **Q2. Generalization Beyond Base-to-Instruct Direction**
>
> We conduct an additional experiment where we merge SpeechCombine with a medical domain model found in Huggingface (Intelligent-Internet/II-Medical-8B) to see if the medical domain knowledge can be transferred to SpeechCombine. Unfortunately, the experiments are still running due to the tremendous volume of experiments. We will include the results in our next round of response.

---

> > ### Author Rebuttal · Reviewer_rv6E · 2026-04-01
> >
> > Thanks for the response and the follow-up experiments. I would like to raise my score from 3 to 3.5.

---

> > > ### Author Response · Authors · 2026-04-06
> > >
> > > As promised, we would like to provide the experimental results on **generalization to other weight directions**. Specifically, we obtain a weight difference medical model found in Huggingface (Intelligent-Internet/II-Medical-8B), and compute its weight difference from the instruct model ($\Delta \theta_{med} = \theta_{med} - \theta_{inst}$). We then perform weight merging as follows
> > > $$
> > > \begin{aligned}
> > > \theta_{SCmed} &= \theta_{based} + \Delta \theta_{inst} + \Delta \theta_{med} + \lambda \Delta \theta_{speech} \\
> > > &= \theta_{SC} + \Delta \theta_{med}
> > > \end{aligned}
> > > $$
> > > We convert [MedMCQA](https://medmcqa.github.io/), a medical-domain QA benchmark into a speech QA benchmark by synthesizing the question into speech. We then test the SLMs on the converted benchmark. The accuracy results (\%) are as follows:
> > >
> > > | SpeechCombine | SpeechCombine-Med | ASR + II-Medical |
> > > |----|----|----|
> > > | 48.41 | 66.79 | 69.46 |
> > >
> > > As can be observed, merging the medical weight difference successfully brings the performance of SpeechCombine up by ~20 percentage points, almost as high as the ASR _ II-Mdical topline. This shows that the SpeechCombine framework not only incorporates general instruction-following capabilities, but other domain-specific capabilities as well.
> > >
> > > ----
> > >
> > > Meanwhile, we noticed that you decided to raise the score by a **0.5 increment**, which is an unconventional increment.
> > >
> > > If this is because you were waiting for the pending results, then we hope that the results provided above could address the remaining concern and lead to a full point increment.
> > >
> > > If this is because of something else, then we would like to bring to your attention that in addition to the merits you already observed in SpeechCombine (**clean concept, broad experimental coverage, outperforming direct baselines, generalizability** etc.), our new experiments also demonstrate some **other encouraging capabilities of SpeechCombine**. For example -
> > > * SpeechCombine outperforms many **new baselines**, including Qwen2.5-Omni, Audio Flamingo 3, and Inserter (See **response to RzSR, W2**)
> > > * We have performed further evaluation where we remove the speech vector and instruction vector to validate their roles (See **Response to RzSR, W5** and **RhVK, W2**)
> > >
> > > We hope these new merits can provide further justifications to **round up** your final score, instead of rounding down.

---

### Decision · Program_Chairs · 2026-04-30

**Decision:**

Accept (regular)

**Comment:**

The paper proposes SpeechCombine, a lightweight approach to building an instruction-following speech language model (SLM) without explicit speech instruction tuning. The authors achieve this by computing two weight-space directions: a speech adaptation vector (from continuous pre-training on 30k hours of speech) and an instruction-following vector (the difference between base and instruct versions of a text LLM). These are combined to yield a model that retains text instruction-following skills while operating on speech inputs.

Following the rebuttal, Reviewer RhVK found their soundness concerns adequately addressed and is supportive of the paper. Reviewer rv6E acknowledged the improvements but only marginally raised their score (to 3.5). Reviewers TTCr and RzSR maintained their negative scores (3). RzSR explicitly noted that while the empirical gaps were patched, fundamental concerns remain regarding limited novelty (similarity to existing task arithmetic) and whether the pipeline truly disentangles speech reasoning from text anchoring. Reviewer TTCr maintained a negative score without providing substantive post-rebuttal justification, which led to an escalation from the authors.

While the scores lean negative, the authors have objectively satisfied the empirical requests made by the reviewers in the initial phase. The methodology, while relying heavily on known task arithmetic principles, provides a highly efficient and effective alternative to expensive large-scale speech instruction tuning. Given the strong empirical defense mounted by the authors during the rebuttal phase, the paper presents a solid contribution worthy of discussion, meriting a Weak Accept.